# Previously hidden landslide processes revealed using distributed acoustic sensing with nanostrain-rate sensitivity

Susanne M. Ouellet [1] ✉, Jan Dettmer [1], Matthew J. Lato[2], Steve Cole[3], D. Jean Hutchinson [4], Martin Karrenbach [5], Ben Dashwood[6], Jonathan E. Chambers[6] & Roger Crickmore[7]

Landslides sometimes creep for decades before undergoing runaway acceleration and catastrophic failure. Observing and monitoring the evolution of strain in time and space is crucial to understand landslide processes, including the transition from slow to fast movement. However, the limited spatial or temporal resolution of existing landslide monitoring instrumentation limits the study of these processes. We employ distributed acoustic sensing strain data below 1 Hertz frequency during a three-day rainfall at the Hollin Hill landslide and quantify strain-rate changes at meter and sub-minute scales. We observe near-surface strain onset at the head scarp, strain acceleration at a developing rupture zone, retrogression towards the scarp, and flow-lobe activity. These processes with displacements of less than 0.5 mm are undetected using other methods. However, the millimeter processes over three days agree with previously observed seasonal landslide patterns. Here, we show landslide processes occurring with nanostrain-rate sensitivity at spatiotemporal resolution previously not possible.

Landslides are a major global geohazard, resulting in the loss of thousands of lives[1] and an estimated average of 20 billion USD in economic losses annually[2]. Slow-moving landslides are associated with movement rates ranging from millimeters to several meters per year[3]. However, these landslides may also accelerate without apparent warning[3,4]. These points highlight the unpredictability of slow-moving landslides and the importance of detecting the early onset of motion and understanding their movement patterns. Furthermore, landslide risk is expected to increase due to anthropogenic factors, including urbanization, deforestation, and the effects of climate change (e.g., increased precipitation, permafrost degradation, and wildfires)[3,5–8]. As settlements and infrastructure increasingly encroach upon areas with potential landslide hazards, effective monitoring systems are a critical component to support informed decisions for mitigation of landslide risk.

Landslide monitoring systems encompass a range of sensors and techniques to detect changes at the Earth's surface (e.g., geodetic surveying, tiltmeters, and remote sensing) and at depth (e.g., inclinometers, acoustic emissions, and piezometers)[9–13]. Geophysical methods support landslide monitoring by inferring changes occurring at depth over broad spatial areas[14]. Satellite and ground-based remote sensing technologies excel in providing broad spatial coverage and enabling landslide monitoring at sites with difficult or hazardous access. However, they are limited in capturing sudden acceleration over shorter timescales and where seasonal effects may impede measurements[3,15,16]. In comparison, distributed fiber optic sensing technologies provide broad spatial coverage, but usually require more installation efforts versus remote sensing. Once installed, distributed fiber optic sensing can provide ongoing near real-time measurements[17,18]. Their combined high spatial and

[1]Department of Earth, Energy and Environment, University of Calgary, Calgary, AB, Canada. [2]BGC Engineering, Ottawa, ON, Canada. [3]OptaSense, Chino, CA, USA. [4]Queen's University, Kingston, ON, Canada. [5]Seismics Unusual Ltd, Brea, CA, USA. [6]British Geological Survey, Keyworth, Nottingham, UK. [7]OptaSense, Farnborough, UK. ✉e-mail: susanne.ouellet2@ucalgary.ca

temporal resolution makes them an attractive option to consider for landslide monitoring applications[19,20].

Distributed acoustic sensing (DAS), also referred to as phase-sensitive optical time-domain reflectometry, is a fiber optic sensing technology relying on the phenomena of Rayleigh backscattering and is sensitive to axial strain and temperature perturbations in the fiber[19]. By injecting pulses of coherent laser light into an optical fiber, an optical phase change is recorded, resulting from the backscattered light between two sections of fiber[21]. As DAS permits a fiber optic cable to be repurposed into an array of broadband seismic sensors, it has effectively spurred a burgeoning field of research known as fiber-optic seismology[22–28]. DAS capabilities to monitor changes at low frequencies (i.e., toward 0 Hz) are less studied. Low-frequency DAS was used to characterize hydraulic fracturing geometry, demonstrating the value in the low-frequency domain[29]. Other studies, referring to a DC-coupled DAS as a distributed Rayleigh sensing system, demonstrate the capability of low-frequency DAS to measure changes in strain associated with the movement of slow-moving shallow landslides[30,31].

We reveal the kinematics of a slow-moving landslide at the Hollin Hill Landslide Observatory in England by employing low-frequency (<1 Hz) DAS data over a 3-day period. The data are acquired with nanostrain-rate sensitivity, 1 Hz temporal sampling, and a 4-m spatial resolution over 925 m of optical fiber. This method provides insights into rainfall-driven near-surface landslide motion and quantifies the spatiotemporal landslide sequence, including the onset of slope movement, retrogression of the landslide and flow-lobe activity near the toe. The processes of this sequence occur at timescales of minutes and exhibit spatial patterns of meter scale. We compare the results with collocated geotechnical instrumentation to demonstrate the unparalleled spatiotemporal resolution of low-frequency DAS. Existing monitoring methods cannot resolve the observations we present. For example, monitoring methods such as ground-based interferometry and automated inclinometers (i.e., ShapeArrays) can resolve up to sub-millimeter displacements over minute timescales, but here we resolve changes in the order of nanometers with 1 Hz sampling frequency[15,32].

The Hollin Hill Landslide Observatory, located in North Yorkshire, UK[32], is one of the most studied slow-moving landslides in the world and has been monitored by the British Geological Survey since 2008[3,33]. The landslide is classified as a very slow to slow-moving composite multiple earth slide–earth flow[9,34], with average velocities typically ranging from 0.5 to 3.5 m/year[9,35]. Earlier studies[9,31,32,35–37] divide the landslide into three major domains: the rotation-dominant domain (above mid-slope), the translation-dominant domain (mid-slope), and the flow-dominant domain (below mid-slope). Slope failure is due to the presence of very weak and highly weathered mud rocks within the landslide-prone Whitby Mudstone Formation (WMF), which outcrops on the valley side. Translational deformation of the WMF leads to rotational deformation above from unloading and oversteepening of the toe along the upper slope. The progressive displacement of the translational deformation occurs as a flow-like behavior over the Staithes Sandstone Formation (SSF) below the mid-section[9]. Landslide processes at Hollin Hill have been studied with multiple site characterization and monitoring methods, including seismic refraction tomography, geoelectrical resistivity, self-potential, inclinometers, piezometers, lidar change detection, interferometric synthetic aperture radar (inSAR), and cone penetration testing[9,32,33,35–38]. Earlier studies integrated multiple methods to reveal different controls on landslide movement. For example, 3D time-lapse imaging of inferred slope moisture content (from inverted resistivity models) demonstrate moisture accumulation in the upper slope over wintertime correlating with landslide reactivation and drainage from the mudstone to sandstone formation in the summertime[19,35].

A single-mode fiber-optic cable of 925 m length is buried ~10 cm below the ground surface at the site (Fig. 1). Most of the fiber-optic cable installed is of tight-buffered construction, but 140 m of loose-tube, gel-filled cable was placed alongside a similar length of the tight-buffered cable for comparison. Our analysis focuses exclusively on data acquired from the tight-buffered cable due to its improved strain transfer properties. The British Geological Survey acquired data along the entire cable length using an OptaSense ODH-F interrogator unit with a spatial sampling interval of 1 m (i.e., channel) and a spatial resolution of 4 m (i.e., gauge length) over a three-day period in January 2021. The interrogator unit was housed in a nearby barn, located ~750 m from the site. The cable was buried in a shallow trench using a small backhoe and shovel, running from the interrogator housing along a farm track to the site[30]. Along the farm track, the cable was encased in protective housing and was laid bare at the site for improved coupling with the surrounding formation (Fig. S1). The processed cable sections (Fig. 1a) are parallel to the direction of slope movement covering an area of ~135 m by 50 m. The approximate slope gradient in the instrumented area is 15 degrees. Collocated instrumentation includes piezometers, automated inclinometers (ShapeArrays), and a weather station (rain gauge, air temperature, barometric pressure)[9]. The locations of the fiber optic installation and collocated instrumentation are shown in Fig. 1. Terrestrial lidar data were acquired in November 2020.

Strain perturbations along the cable result in an optical phase change $\Delta\phi$ of the Rayleigh-backscattered light over the gauge length ($L_G$) of the measurement, which is mapped along the axis of the fiber using principles of optical time-domain reflectometry, and has a linear relationship with strain $\varepsilon$ [19,20]

$$\varepsilon = \frac{\lambda \Delta\phi}{4\pi \zeta n L_G},\tag{1}$$

where $n$ corresponds to the refractive index of the fiber, $\lambda$ corresponds to the wavelength of the coherent laser pulse and $\zeta$ corresponds to a scalar multiplicative factor to account for changes in the index of refraction (Methods). The optical phase changes and corresponding strain measurements represent a change from a baseline measurement occurring at the start of data acquisition. All references to strain $\varepsilon$ and strain-rate $\dot{\varepsilon}$ herein represent a relative measurement from the baseline measurement on January 12, 2021 at 11:38 UTC.

In this work, we apply a 1-Hz low-pass filter, then convert the optical phase data to units of strain and strain rate. To support interpretation, we create spatiotemporal images of the cable strain-rate data along select cable sections extending from the slope crest to toe. Cable sections are numbered from west to east in increasing order. The resulting spatiotemporal images of strain rate reveal complex patterns of accelerating and decelerating strain at the site. Landslide displacements are inferred from strain and velocities are inferred from strain-rate (Methods). These are quantitatively compared with collocated geotechnical instrumentation. We characterize landslide features at the Hollin Hill site by identifying distinct patterns in the spatiotemporal images, including retrogression towards the scarp and a flow surge event near the toe. Our results highlight landslide processes at scales previously unresolved, enabling insights into the kinematic evolution of slow-moving landslides.

## Results
With 1-min temporal resolution (based on the median filter, see Methods), 4-m spatial resolution (akin to the gauge length), and nanostrain-rate sensitivity, we develop strain-rate spatiotemporal images (Fig. 2) with an unprecedented level of detail to assess evolving changes in the near-surface compared to remote sensing methods[15]. Our results highlight the strain-rate patterns observed at the four easternmost cable sections. Spatiotemporal strain-rate images for all six cable sections are included in Fig S2. Strain spatiotemporal images are also available for comparison (Fig. S3).

Positive and negative strain and strain-rate observations correspond to cable extension and compression, respectively, and we use

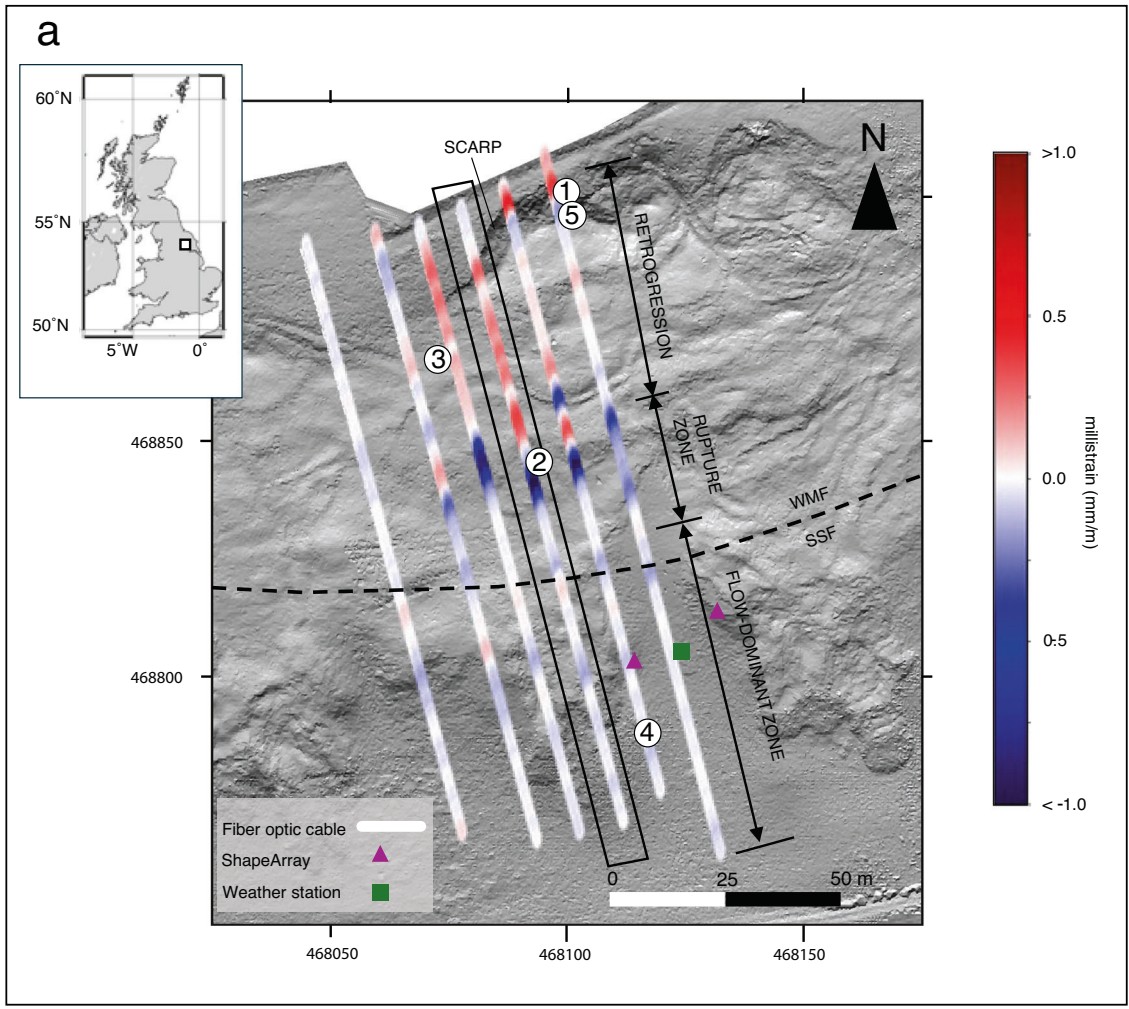

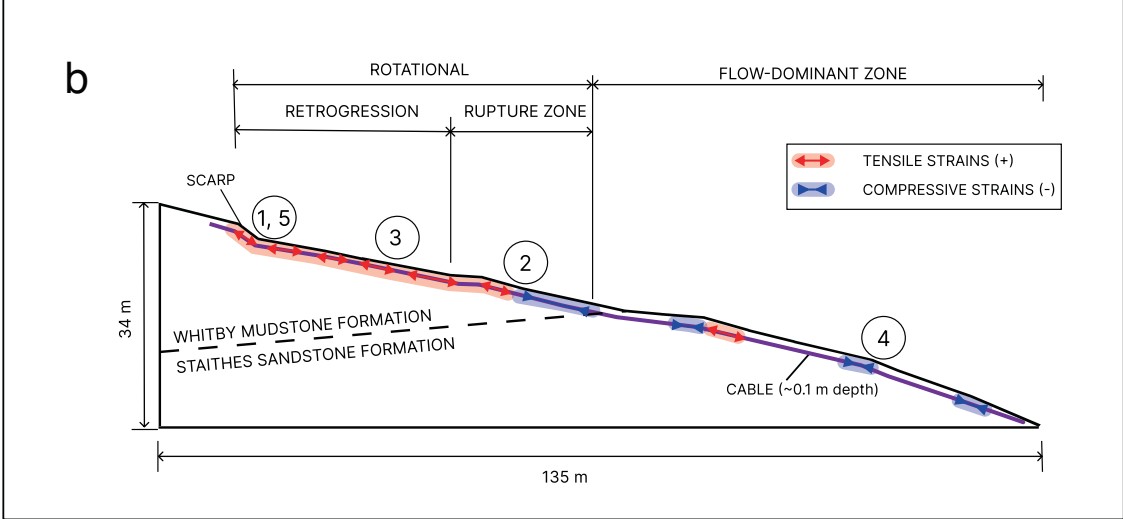

**Fig. 1 | Hollin Hill landslide observatory and the fiber optic cable. a** Inset figure shows the location of Hollin Hill. Inset figure was created using PyGMT[71], a Python wrapper for Generic Mapping Tools[72]. Base image from bare-earth lidar from a November 2020 terrestrial lidar scan. Grid coordinates per the British Grid OSGB36 datum. The in situ Whitby Mudstone Formation (WMF) and Staithes Sandstone Formation (SSF) geological boundaries are per[37]. Unprocessed cable sections perpendicular to the direction of slope movement are not shown. DAS strain data corresponding to 2021-01-15T11:00. Soil moisture content sensor and precipitation gauge are located at the weather station. Numbered sequences show the approximate location of DAS channels, described in the results. **b** Conceptual model illustrating the main landslide zones and approximate strain changes for the same time as in (**a**). Strain vector lengths are qualitative and do not represent actual strain magnitudes. Numbered sequences show the approximate location of DAS channels, described in the results.

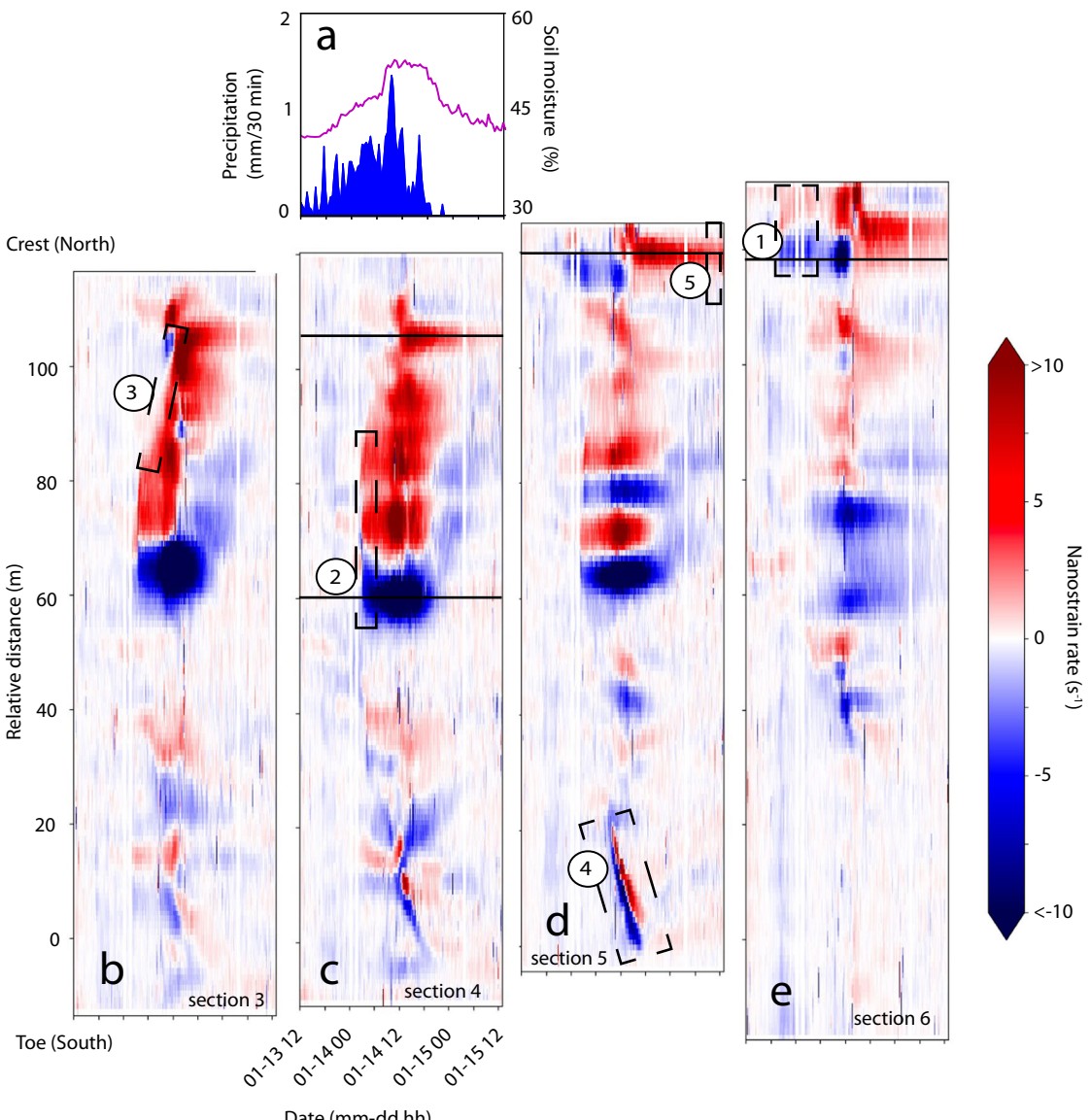

**Fig. 2 | Strain-rate spatiotemporal images.** Strain-rate image positions are adjusted for similar latitude (y-axis) across all four easternmost cable sections (cable sections are shown from west to east, sections are numbered as 3 through 6) from as-built survey of cable. **a** Precipitation (blue) and soil moisture (purple) data. **b** Strain-rate for cable section 3, highlighting Sequence 3. **c** Strain-rate for cable section 4, highlighting Sequence 2 and DAS channel locations featured in Fig.4c (lower solid line) and Fig.4d (upper solid line). **d** Strain-rate for cable section 5, highlighting Sequences 4 and 5, and DAS channel location featured in Fig. 4e. **e** Strain-rate for cable section 6, highlighting Sequence 1 and DAS channel location featured in Fig. 4b.

the extensional or compressional descriptors herein. Thanks to the spatiotemporal characteristics of the DAS data, velocity features of the landslide are obtained per the following approaches: (1) the velocity occurring at a single location (i.e., DAS channel), (2) the velocity of a strain or strain-rate front propagating over multiple DAS channels with time. We guide our analysis using a conceptual framework to interpret key strain and strain-rate patterns, as follows: (1) Extensional strain-rate observations that are upslope of compressional strain-rate occurring over the same time period are interpreted as a shallow slip surface intersecting with the cable (Fig. 3a). (2) Propagation of the extensional strain-rate processes upslope with time are interpreted as slope retrogression. We characterize the retrogressive behavior by analyzing the slope of the strain and strain-rate fronts with time (Fig. 3b). (3) A paired extensional and compressional strain-rate observation propagating downslope is interpreted as a surge of saturated materials (i.e., a flow surge) propagating over the cable (Fig. 3c). (4) A sequence of paired

extensional and compressional strain-rate observations are attributed to topographic and material strength variations (Fig. 3d).

Rainfall occurred over a 1.5-day period (January 13 at noon to January 14 at 18:30) with a cumulative precipitation of 28 mm. Following our conceptual framework of strain-rate patterns (Fig. 3), we interpret the five event sequences which are ordered chronologically: (1) initiation of strain at the head scarp, (2) subsequent triggering of a rupture zone, (3) retrogressive strain towards the scarp, (4) a flow-lobe surge near the toe, and (5) stabilization of the rupture zone with a gradual increase in strain at the scarp. Sequences 1 through 5 are numbered and highlighted by dashed rectangles in Fig. 2b–e. The unparalleled spatiotemporal resolution of our findings reveals complex patterns of landslide movement at previously unresolved scales.

**Strain initiation at head scarp (sequence 1)**
After about 8 h of rainfall, the onset of strain is observed at the northeast corner of the cable at the scarp (Fig. 4b, f). Over the subsequent

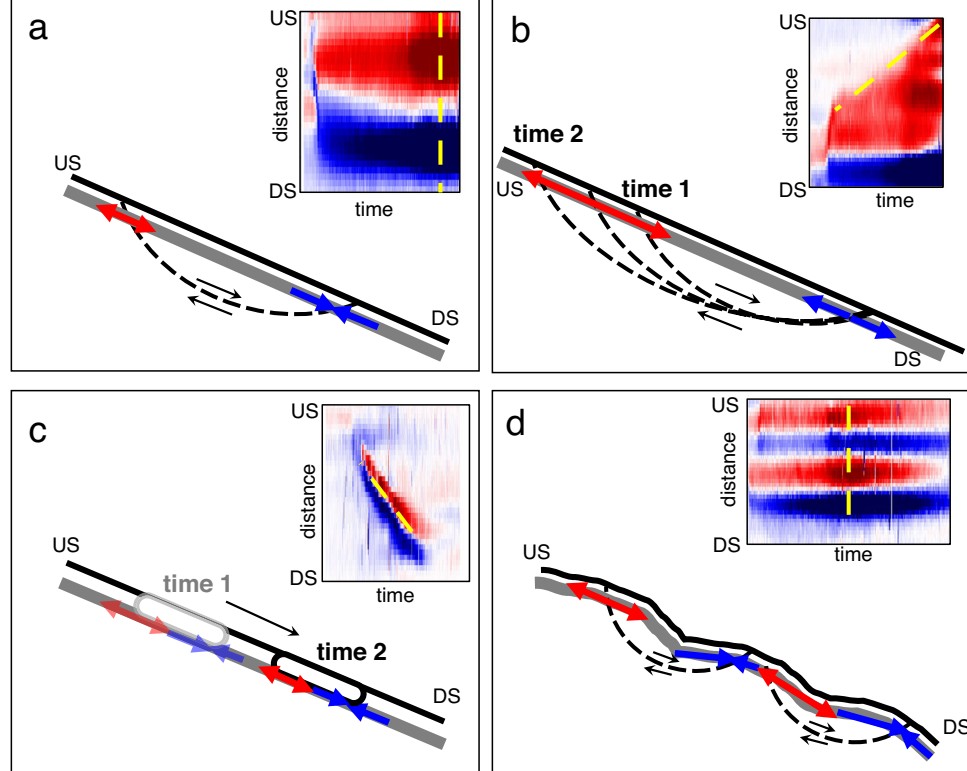

**Fig. 3 | Conceptual framework to support interpretation of strain-rate patterns.** Inset figures represent strain-rate spatiotemporal images selected over select channels and time periods to illustrate key patterns. Fiber optic cable is shown in gray. The red and blue arrows represent extensional and compressional strain-rate observations, respectively. US upslope, DS downslope. The dotted yellow line demonstrates whether inferences are made over a single time period along a series of channels or over multiple time periods along a series of channels. **a** Inferred slip surface. **b** Inferred retrogression. **c** Inferred flow lobe surge. **d** Inferred slip surfaces due to topographic variations.

eight-hour period, we observed a strain rate of ~−4 nm m$^{-1}$ s$^{-1}$ near the main scarp (corresponding to an average inferred velocity of −1 mm per day). The observed pattern at the scarp over this time corresponds to a low amplitude slip surface pattern (Fig. 2e). The strain change correlates with a steady increase in soil moisture content from 43 to 47%.

**Rupture zone triggering (sequence 2)**
On January 14 at ~3:00 UTC, a slip surface pattern develops ~30 m southwest of the main scarp (Fig. 2c). Based on the greater magnitude of the observed strain changes at this location during this time compared to other cable locations, we interpret this as the triggering of a rupture zone. The strain rate decreases to a minimum value of −35 nm/m·s (inferred velocity of −12 mm/day) at 12:00 on January 14 (Fig. 4c). Over this period, the soil moisture content increases from 47% to a maximum of 53%.

**Retrogression towards scarp (sequence 3)**
From the triggering of the rupture zone (January 14 at ~3:00) to ~12 h later, the strain-rate front propagates from the rupture zone upslope towards the scarp (over a ~20 m distance; Fig. 2b). We interpret this as a retrogressive deformation of the landslide. This coincides with accelerating strain near the main scarp (Fig. 4d, 2c, upper solid line).

**Flow lobe surge (sequence 4)**
Further downslope, propagation of the strain-rate front at the flow lobe occurs across ~10 m in the downslope direction over a 4-h period (average velocity of the strain-rate front propagation is ~2.5 m/h; Fig. 2d and Fig. S4). This is interpreted as a surge of superficial material flowing over the cable (not representative of the motion of the rupture zone assessed above, where the material is assumed to be well coupled with the cable, see Discussion).

**Stabilization of rupture zone (sequence 5)**
The final sequence coincides with a stabilization of the rupture zone, where the strain rate approaches near-zero values (Fig. 2d, 4e). The strain continues to increase at more gradual levels at the main scarp up until the end of the DAS acquisition period (Fig. 4b). The soil moisture content decreases to 43%.

Further to the above, historical aerial photography shows the westward propagation of the landslide and the development of the main scarp, occurring following major movement in 2016[38]. With data acquisition over only a 3-day period, our results show that the landslide is continuing to propagate westward, considering the maximum strain changes in the rupture zone to the southwest of the existing scarp (Fig. 5).

**DAS-derived displacements with geotechnical instrumentation**
We rely on collocated geotechnical instrumentation to evaluate the inferred displacements from the DAS strain. Our comparison provides evidence for the reliability of the relative magnitudes of the DAS strain-rate, strain and inferred displacements. Furthermore, it illustrates the low noise levels at a single DAS channel in comparison with the ShapeArray, permitting unprecedented resolution of changes in movement to be detected.

We compare relative displacement data from two vertical ShapeArrays installed near the fiber optic cable to a depth of 2.5 m, at the west flow lobe and east flow lobe in 2013 (Fig. 6). The near-surface (~0.3-m depth) ShapeArray data were processed to obtain relative displacement from a common baseline for comparison with the

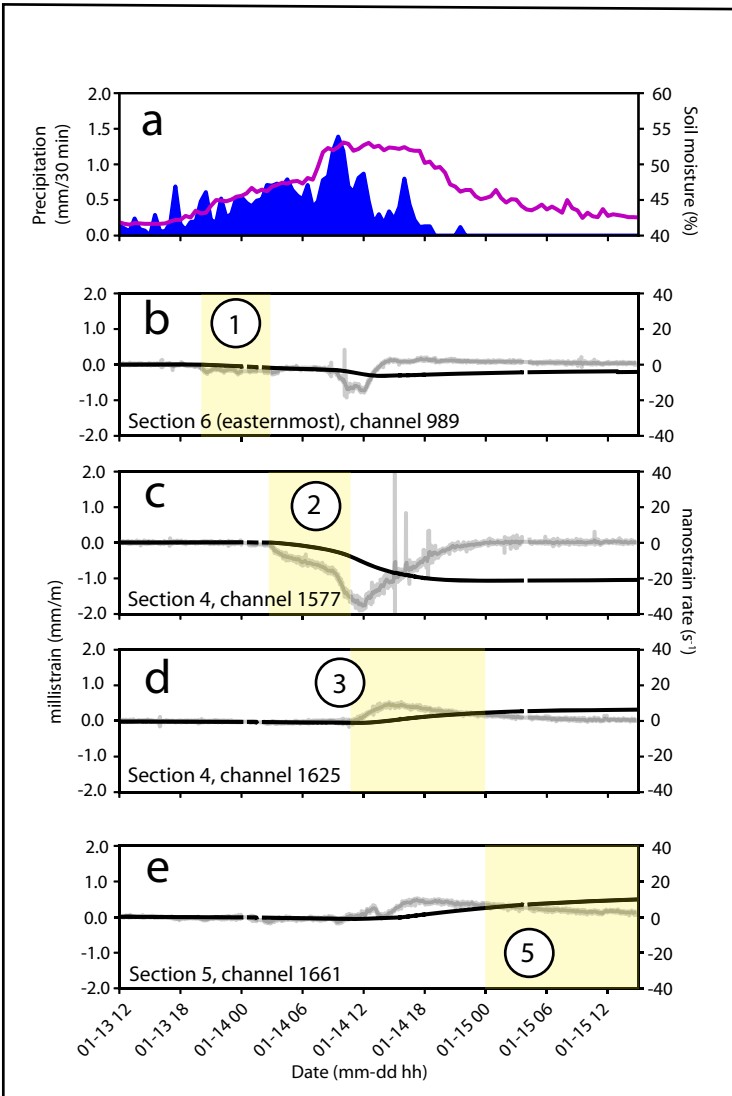

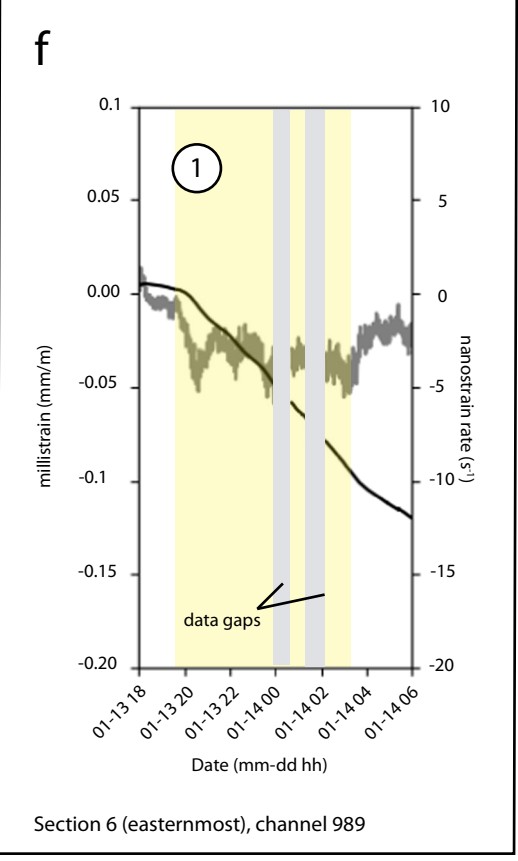

**Fig. 4 | Changes in strain and strain-rate alongside precipitation and soil moisture, with highlighted time periods of interest (sequences 1, 2, 3, and 5).** **a** Soil moisture content (purple) and precipitation data (blue). **b** Strain and strain-rate for DAS channel 989 at cable section 6 (easternmost). Yellow shading highlights Sequence 1. **c** Strain and strain-rate for DAS channel 1577 at cable section 4. Yellow shading highlights Sequence 2. **d** Strain and strain-rate for DAS channel 1625 at section 4. Yellow shading highlights increasing strain at the scarp. **e** Strain and strain-rate for DAS channel 1661 at cable section 5. Yellow shading highlights Sequence 5. **f** Inset of Fig. 4b to show the initial onset of strain observed at DAS channel 989, cable 6. Strain data is shown as black solid line and strain rate data is shown as gray solid line data. Approximate channel locations are indicated in Figs. 1, 2 with sequence numbering.

inferred DAS absolute displacements (Methods). Generally, higher magnitudes of both ShapeArray and DAS data are observed at the west lobe. The total displacement at the east-lobe and west-lobe ShapeArrays are 0.6 and 0.8 mm, respectively. The DAS-derived displacements at nearby locations vary from 0.3 mm (between east and west lobes) to 0.4 mm (adjacent to west-lobe ShapeArray). The observed discrepancies are likely a result of (1) the accuracy and precision of the ShapeArray data at very small (<1 mm) displacements; (2) the strain transfer occurring inside the fiber-optic cable and the coupling of the cable to the surrounding ground; (3) the different depths of measurement and sensor locations; and (4) the different nature of the measurements, where the ShapeArray data provide a discrete measurement at a sensor and the DAS data provide a distributed measurement over a 4-m length of optical fiber (i.e., the gauge length; $L_G$). The observed variability between subsequent ShapeArray samples (±-0.1 mm) illustrates higher noise in comparison with the DAS estimates (Fig. 6). As the strain is enclosed within a fiber-optic cable comprising multiple layers with imperfect coupling to the surrounding

ground, this results in a lower strain in the core versus the strain in the surrounding ground[39,40]. The temporal changes in displacement are well correlated, with the main increase in strain observed to occur on January 14 at noon in both the ShapeArray and DAS data.

## Discussion

We observe landslide processes with previously unresolved spatio-temporal resolution and demonstrate how these processes mimic those occurring over longer (i.e., seasonal) timescales. Our results are supported by findings from earlier studies describing similar movement patterns that demonstrate the influence of seasonal precipitation and soil moisture content on slope movement[35,36,38,41]. This agreement with earlier studies supports the validity of our findings and suggests that the processes governing landslide behavior are scale-invariant (i.e., similar regardless of the scale at which they are observed). For example, a past study hypothesized that deforming materials of different relative densities may contribute to a deformation wave progressing through the slope[41] (i.e., a flow surge event). Our study

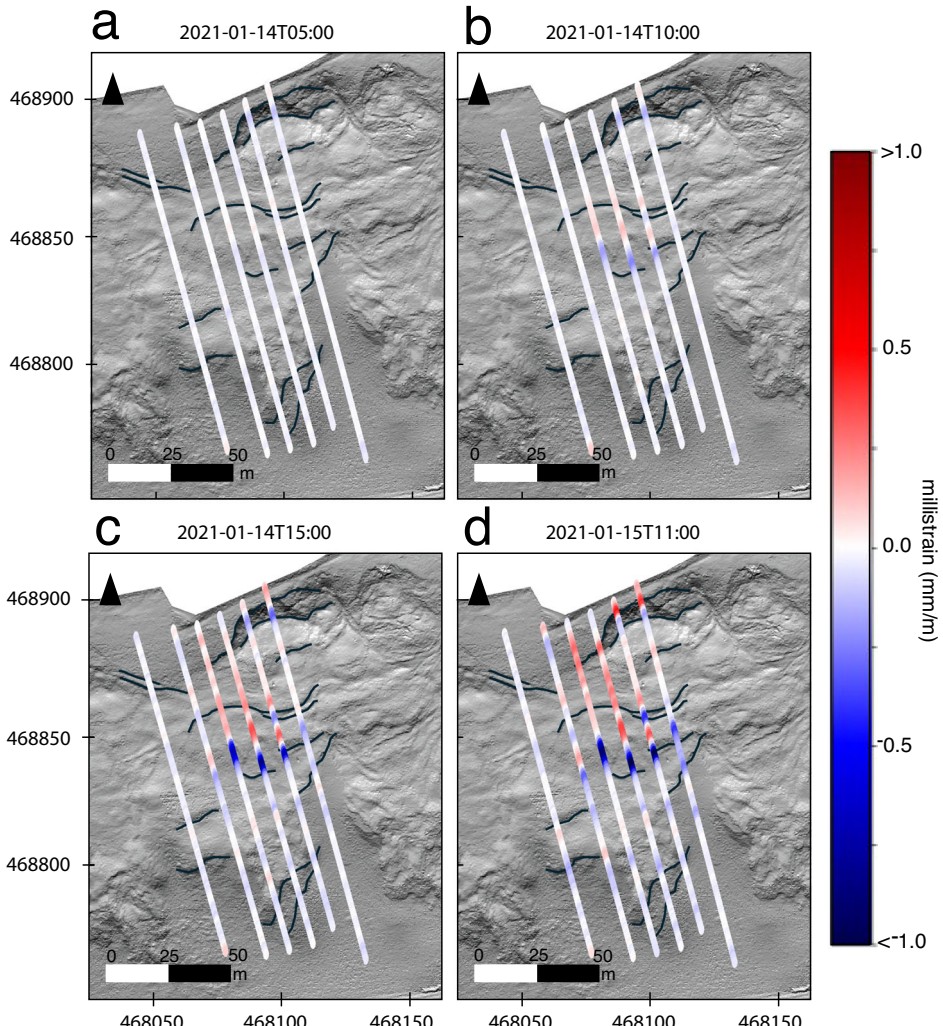

**Fig. 5 | Change in DAS strain at key time periods.** Change in strain to illustrate visible changes in strain recorded from DAS data overlain onto lidar bare-earth imagery from November 2020. Grid coordinates per the British Grid OSGB36 datum. Topographical features near the cable are accentuated in black based on visual inspection of lidar imagery. **a** DAS strain on January 14, 2021 at 05:00. **b** DAS strain on January 14, 2021 at 10:00. **c** DAS strain on January 14, 2021 at 15:00. **d** DAS strain on January 15, 2021 at 11:00.

provides observations of strain-rate front propagation in the flow-dominant zone to characterize the spatial extents of an interpreted flow surge event occurring over an ~8-hour period, along with its associated strain-rate front propagation velocity of 2.5 m/h (Fig. S4). Our method can be used to distinguish kinematic zones which vary over time to support landslide hazard assessments (e.g., discerning the motion characteristics of slower soil-creep events from more rapid flow-surge events). This capability enables landslide monitoring on spatiotemporal scales that are currently poorly understood and improves our ability to characterize landslide events and behaviors using current classification systems[42]. Importantly, this addresses existing limitations of current deformation monitoring techniques based on state-of-the-art remote sensing technologies such as ground-based interferometry and Doppler radar[15] which lack temporal resolution combined with sensitivity to small displacements. Therefore, our DAS-based method can provide crucial information for landslide early-warning applications by enabling monitoring over broad temporal and spatial scales[3,15,43].

The initial strain change is observed at the northeast corner at the location of the main scarp. This is likely a result of the main scarp providing a direct rainfall-infiltration pathway to the near-surface cable. Similar to the observations described by ref. 44, rainfall-induced

saturation of the soil at the depth of the cable is believed to decrease the friction between the cable and surrounding ground, resulting in a small decrease in strain. Although the strain-rate changes show visible activity at the scarp, the overall change in strain in the northeast corner remains negligible in this early period in comparison with the maximum strain changes occurring after sustained rainfall. This highlights a major advantage of using strain-rate data alongside strain to help distinguish the onset of changes[3,15,43].

Following the onset of strain, the development of a rupture zone to the south of the scarp becomes apparent. We do not observe a continuous propagation of strain between the location of the scarp and the initial zone of rupture (Fig. 5b). This is likely a result of our interpreted slip-surface geometry at this location, where the greatest observed tensile strains at the near-surface cable are expected to occur where the slip surface intersects with the cable (Fig. 3a). We observe compressive strains at the toe of the interpreted rupture zone, followed by tensile strain at the scarp, suggesting retrogression of the slope (Fig. 2b). As the slide retrogresses towards the scarp, the strain between these two locations increases as expected with our conceptual model (Fig. 5c). The tensile and compressive strains at the main rupture zone indicate likely slip surface entry and exit points. Additional observed strain-rate patterns are likely impacted by a

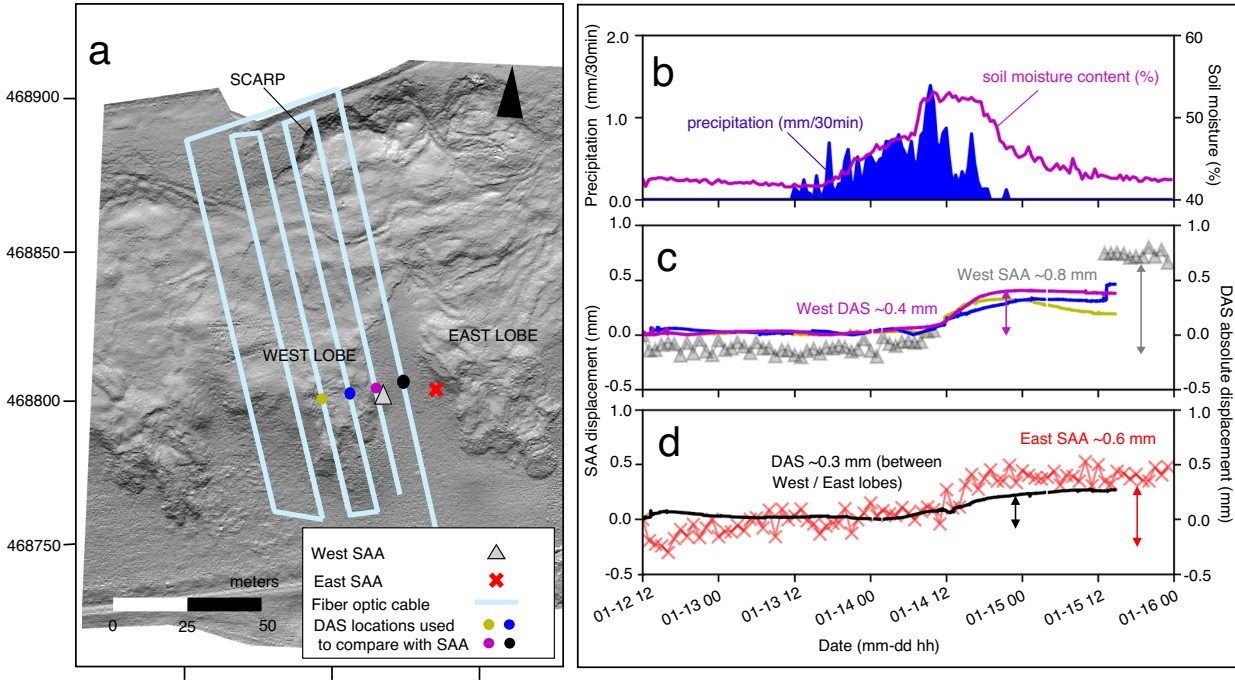

**Fig. 6 | Comparison of ShapeArray displacements with DAS-derived displacements. a** Overview figure indicating the approximate locations of DAS channels selected for comparison alongside the location of the two ShapeArrays (SAA), depicted as a gray triangle and red cross on the west and east lobes, respectively. The west lobe ShapeArray did not record data from January 14 at 11:00 to January 15 at 13:00 for reasons unknown. Lidar bare-earth imagery data were acquired in

November 2020. Grid coordinates per the British Grid OSGB36 datum. **b** Precipitation and soil moisture data. **c** Time displacement plot comparing the DAS-derived displacements with the west lobe ShapeArray displacements. **d** Time displacement plot comparing the DAS-derived displacements with the east lobe ShapeArray displacements.

combination of factors, including slope topography, strain redistribution from the shallow landslide movements, and variations in material parameters (Fig. 3d and Fig. S5). Overall, the distribution of positive and negative strains aligns with our interpretation of the main landslide processes.

These complex spatiotemporal patterns represent an important insight into shallow landslide kinematic behavior at scales not observed previously. The spatiotemporal DAS images and geotechnical data support our interpretation of the sequential occurrence of events and active areas of movement. The DAS data reveal changes in displacement of <1 mm that correlate with ShapeArray and Lidar bare-earth imagery data, further highlighting how DAS data can complement slope monitoring networks. However, as our dataset consists of a three-day period, a longer data acquisition period is required to understand how the strain-rate patterns evolve over longer time periods (i.e., months to years) (Fig. S6).

Distributed fiber optic sensing (DFOS) technologies, encompassing distributed acoustic, strain, and temperature sensing, are primarily sensitive to changes occurring along the axial direction of the optical fiber. As such, strain changes along a slip surface at depth (where the direction of movement is not aligned with the cable) could be masked. This raises a significant consideration for the cable installation geometry in future experiments. Ideally, the cable should be aligned for maximum sensitivity with changes (parallel to the direction of slope movement). Additional cable geometries, orthogonal to the direction of slope movement, could also be used to support the characterization of multi-directional landslide movement patterns. For rotational failure surfaces, the DAS sensitivity to strain changes will vary depending on the difference in angle between the slip surface and the cable. However, others[40,45] have shown that DFOS can still be used successfully to detect and monitor the development of landslide shear zones over time, even in situations where the shear zone is perpendicular to

the cable (e.g., when the cable is installed down a borehole). Future research should investigate the effect of cable geometry on the detectable strain thresholds and on the accuracy of quantifying deformation. For example, acquiring DAS data from a downhole fiber optic cable paired with an inclinometer or ShapeArray, combined with a trench installation, would provide useful information on the capabilities of DAS to capture deformations occurring at depth. Regardless, our experiment demonstrates the valuable insights gleaned from a near-surface trench installation.

We assume the cable is relatively well coupled to the ground for the entirety of the acquisition period, except for the flow-lobe activity where superficial material flows over the cable (Results and Fig. S4). A laboratory-scale failure with a distributed strain sensing technology demonstrates the different phases of strain detection[44]. The authors highlight a period of partial coupling followed by full decoupling to illustrate how the strain is no longer representative of the ground strain following decoupling. As the Hollin Hill site does not undergo a major failure (the ~1 mm of deformation is relatively minor in comparison with earlier and subsequent movements, Fig. S6), we believe our assumptions of good coupling between the cable and the surrounding ground over the acquisition period are reasonable. Furthermore, the steady strains observed at most channels support our claim of negligible slippage or decoupling. However, pullout experiments could support future experiments and monitoring activities with DAS by providing an estimate of the maximum strain that can be experienced by the cable prior to likely decoupling, as demonstrated in ref. 44. Our comparison with nearby ShapeArray instrumentation provides us with a simplified approach to estimating the strain transfer between the optical fiber and surrounding formation. In general, the displacement magnitudes from the ShapeArray instrumentation are twice the magnitudes of inferred displacement from the nearby DAS channels. This relationship between ground displacement from ShapeArray instrumentation and DAS

inferred displacement can be extended to estimate a strain transfer of 0.5, under the assumption of similar coupling throughout the length of installed fiber optic cable and with consideration to the different nature of the measurement (i.e., point-based versus distributed; see Results). With only two ShapeArray sensors for comparison, and one of them at a location offset from the flow lobe, further studies on the inferred quantitative DAS-derived displacements are needed to provide a more robust estimate of these assumptions. Further to the above, different types of cable construction will result in different strain transfer coefficients. Our study uses DAS results acquired from a tight-buffered fiber optic cable, providing improved strain transfer from the surrounding formation to the cable. In contrast, a loose-tube (gel-filled) cable type is designed to minimize the strain transferred to the optical fiber from the surrounding formation. As such, this cable type is deemed less effective for monitoring changes in strain[46]. Although our study focuses on strain changes in the near-surface for landslide characterization using DAS, complementary DFOS technologies such as distributed temperature sensing could be considered to provide measurements of temperature changes along the fiber and support the interpretation of DAS data.

As the velocity of slow-moving landslides can vary significantly over both time and space, an effective landslide monitoring system should be capable of capturing both longer-term changes in trend and the potential for accelerating conditions leading to a catastrophic failure over the landslide extents[3,15]. Our work demonstrates the capabilities of DAS in resolving highly sensitive changes in both time and space to uncover landslide processes not previously known. Landslide forecasting methods commonly rely on displacement monitoring as the primary indicator to inform the time of failure[43,47–49]. Although forecasting time to failure was outside the scope of this work, similar methods could also be applied using DAS strain and strain-rate data for landslide monitoring applications.

Over a 3-day period encompassing high-intensity rainfall, we quantify the kinematics of the spatiotemporal landslide sequence from DAS strain and strain-rate changes. Our findings reveal landslide processes, including strain onset, retrogression, and flow-lobe activity, with sub-minute temporal resolution and nanostrain-rate sensitivity. Although this study relies on the low frequencies (<1 Hz) from a near-surface DAS dataset, DAS fiber-optic sensing also provides rich information at higher frequencies[22,23,50–52] which may enable further discoveries about landslides. This opens research avenues to explore for landslide monitoring, by pairing seismic monitoring with the static strain-change monitoring we presented here. Since DAS monitoring of tens of kilometers of fiber is common, the method we presented could effectively complement existing remote sensing techniques and be employed in early-warning systems due to its low computational cost and DAS systems' ability to transmit data in real-time[25–27,53,54]. Considering the increasing frequency of landslides driven by climate change[5,55], our DAS method could provide critical information for slope stability monitoring in densely populated areas.

## Methods

### Technology background

DAS relies on Rayleigh light backscattering, where small-scale variations in the refractive index of the fiber, due to its crystalline structure, allow for Rayleigh backscattering to occur. The phase of the backscattered light is altered where there are localized disturbances to the fiber. By measuring the difference in phase between two sections of the fiber, a signal is obtained that can be converted to a strain or strain rate for output, using the speed of light and known properties of the fiber[22,56]. As such, DAS does not measure absolute strain, but instead measures relative change in strain dependent on the optical phase change attributed to a disturbance in the fiber[21]. The DAS measurement corresponds to a discrete point (i.e., channel) along the fiber, representing an average measurement which is centered over a length of fiber referred to as the gauge length ($L_G$)[51,57] (Fig. 7).

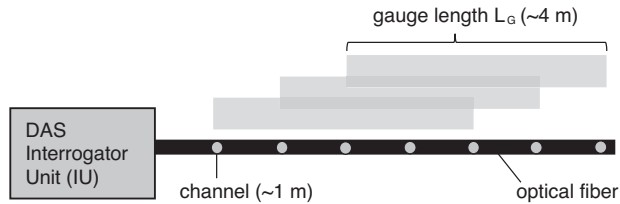

**Fig. 7 | Illustrative diagram of the DAS interrogator unit with the measurement points (channels) and gauge length.** The channel spacing used for this study is ~1 m. The gauge length ($L_G$) used for this study is ~4 m.

The output of DAS is dependent on the type of interrogator employed. For this experiment, the raw DAS output was provided in units of optical phase change ($\Delta\varphi$) and can be linearly related to the average strain ($\varepsilon$) along the axis of the fiber over the gauge length $L_G$ (Eq. 1)[23]. For this study, $n$ corresponds to a refractive index of 1.468 for the single-mode fiber, $L_G$ corresponds to a gauge length of 4.084 m, $\lambda$ corresponds to the wavelength of the coherent laser pulse in a vacuum of 1550.12 nm and $\xi$ corresponds to a scalar multiplicative factor of 0.81, accounting for changes in the index of refraction. Separate from the gauge length, the channel spacing determines the spatial distance between each sample along the optical fiber (Fig. 7). When the channel spacing is less than the $L_G$, the effect of the latter is the same as applying a moving average window of length $L_G$ centered at each channel.

As the output from DAS is represented as an average strain $\varepsilon$ over the gauge length, displacement $d$ is inferred by integrating the average strain over the gauge length $L_G$

$$d = \varepsilon L_G \tag{2}$$

Strain rate is obtained by performing a temporal derivative on the average strain $\varepsilon$

$$\dot{\varepsilon} = \frac{d\varepsilon}{dt} \tag{3}$$

Velocity $v$ is inferred by integrating the average strain rate $\dot{\varepsilon}$ over the gauge length $L_G$

$$v = \dot{\varepsilon} L_G \tag{4}$$

### Temperature effects

In addition to strain changes, temperature changes $\Delta T$ also cause a linear change in the optical phase change $\Delta\varphi$ recorded by DAS[58–60]. This can be represented using the DAS thermal coefficient $C_T$ and the DAS strain coefficient $C_\varepsilon$[58], where

$$\Delta\varphi = C_T \Delta T + C_\varepsilon \varepsilon \tag{5}$$

We estimate the influence of temperature changes on our DAS data by using a nearby temperature sensor installed at the fiber optic cable depth (10 cm below ground surface). The temperature sensor is installed at the location of the weather station (Fig. 1a). We follow the approach described by ref. 58 to compute the equivalent strain for the general temperature change of 1.7 °C (Fig. S7) occurring over our DAS acquisition period. The DAS thermal ($C_T$) and strain ($C_\varepsilon$) coefficients using Eqs. 6 and 7.

$$C_T = \frac{4\pi n L_G}{\lambda} \left( \frac{\xi_T}{n} + \frac{\xi_\varepsilon}{n}\alpha + \alpha \right) \tag{6}$$

**Table 1 | Properties of single-mode optical fiber and DAS data acquisition settings**

| Variable | Symbol | Value | Units | Reference |
|---|---|---|---|---|
| Thermo-optic coefficient | $\xi_T$ | $1.20 \times 10^{-5}$ | 1/°C | 70 |
| Strain-optic coefficient | $\xi_\epsilon$ | −0.32 | 1/strain | 70 |
| Thermal expansion coefficient | $\alpha$ | $8.00 \times 10^{-7}$ | 1/°C | 70 |
| Wavelength | $\lambda$ | 1550.12 | nm | DAS acquisition settings |
| Index of refraction | $n$ | 1.468 | Dimensionless | 70 |
| Gauge length | $L_G$ | 4.08 | meters | DAS acquisition settings |

$$C_\epsilon = \frac{4\pi n L_G}{\lambda}\left(1 + \frac{\xi_\epsilon}{n}\right) \tag{7}$$

The variables used to obtain $C_T$ and $C_\epsilon$ are outlined in Table 1.

Temperature and strain are represented as average values over the gauge length of 4.1 m. The DAS thermal and strain coefficients were computed as 427 radians/°C and $3.8 \times 10^7$ radians/strain. As such, a change in 1 °C corresponds to a change of $1.1 \times 10^{-2}$ millistrain. Over the three-day DAS data acquisition period, the ground temperature decreased by about 1.7 °C (Fig. S7). This change can be converted into an equivalent strain of $1.84 \times 10^{-2}$ millistrain. This value is approximately two orders of magnitude lower than the main strain changes observed over the rainfall period that were used to inform our conceptual model (Fig. S3). As such, we deem it reasonable to neglect the effects of temperature in our work. However, incorporating the effects of temperature change could be used to increase the DAS measurement precision at lower strains.

## Data processing

Although most geotechnical studies involving DAS to date focus on its applications as a seismic sensor[61–63], we focus on low-frequency (<1 Hz) DAS measurements to extract meaningful information on changes in static strains. This was demonstrated by ref. 29, who used low-frequency DAS to measure small changes in strain for hydraulic fracturing applications. Here, we extend the application of low-frequency DAS toward a slow-moving landslide using a tight-buffered fiber optic cable. The following data processing steps are implemented to obtain low-frequency DAS strain, displacement, strain rate, and velocity measurements.

DAS data were acquired using the OptaSense ODH-F interrogator unit in the quantitative mode. The obtained data represent the optical phase change at a sampling rate of 500 Hz. Data were acquired from OptaSense in a proprietary file format of 32 files, with individual file sizes of 10.7 gigabytes. They are decimated to 50 Hz with a low-pass antialiasing filter and exported to HDF5 file format using the OptaSense DxS software, where each file contains all DAS channels over a 1-min period.

The remainder of the data processing steps are completed using the following open-source Python modules: SciPy[64], NumPy[65], Pandas, H5Py, Obspy, Zarr, and Xarray[66]. The processed strain and strain-rate DAS datasets are saved using Xarrays containing appropriate metadata (time, channel, and coordinates) in NetCDF files corresponding to individual cable segments (six segments numbered in increasing order from west to east) with explanatory Jupyter notebooks detailing the following processing steps[67]:

1. The DAS data were decimated to 1 Hz following the application of a low-pass antialiasing filter.
2. Following decimation, the data arrays are reshaped, concatenating the one-minute files to incorporate the full data acquisition time period (~3 days).
3. The concatenated optical phase data are converted to strain using Eq. 1.
4. Strain measurements are obtained by subtracting subsequent samples from the initial sample at time 2021-01-12T11:38:10 for

each DAS channel. No median filter is implemented on the strain data.
5. Strain-rate data were obtained by computing the temporal derivative of strain using the central differences method between consecutive samples (Eq. 3).
6. A 2D median filter is implemented on the strain-rate data, using a window size of three channels and 59 s, with zero padding on data extents.
7. Displacement and velocity data are inferred by integrating the strain and strain-rate data over the gauge length, respectively (Eqs. 2 and 4).

The locations and elevations of the fiber optic cable were collected using a Leica Viva GS14 antenna and CS15 handset at 20-m intervals. Linear interpolation between each as-built location was used to infer location data at each DAS channel.

## Data gaps

Three periods of data gaps, ~45-min duration, are visible in Figs. 2, 4, 6 are due to corrupted data files, where the original data were unable to be recovered. These periods are excluded from the above data processing steps. The data were reindexed to a continuous time vector using the timestamp associated with each filename, and the data gaps were incorporated as "NAN" (i.e., not a number) values.

Geotechnical site data used to support our interpretation are described in the following paragraphs.

## Weather station

A weather station, installed in 2008[9], records soil moisture with a sensor installed at 0.1-m depth and rainfall data recorded at 30-min intervals (Fig. 1). Data from this weather station is publicly available as part of the UK Centre for Ecology and Hydrology[68].

## Displacement measurements (ShapeArray)

ShapeArrays (also known as Shape Acceleration Arrays; SAAs) were installed in 2013 and have been providing near continuous measurements since that time[9]. ShapeArrays consist of an array of microelectromechanical system (MEMS) accelerometers. These instruments measure the acceleration relative to gravity in the x-, y-, and z- directions at rigid segments along the length of the instrument to obtain a tilt measurement at each segment. Using a fixed reference point, relative displacements can be calculated from the tilt data[69]. ShapeArrays were installed to depths of ~2.5 m below ground level with 0.25-m sensor spacing. Instruments were assumed to be installed with the x-direction oriented horizontally parallel to the direction of movement along the downslope axis. In comparison, the DAS represents axial strain along the direction of the cable, parallel to the slope (average slope gradient of 15°). To account for the difference in magnitude of a displacement along the horizontal axis recorded by the ShapeArray and at a slope gradient of 15° recorded by DAS, a correction factor of 0.97 (cos 15°) could be implemented. For this work, we do not implement a correction factor, as 0.97 is negligible for our purposes. However, we note that this should be considered for future

experiments with greater slope gradients comparing DAS and ShapeArray data. Over the DAS acquisition period, the ShapeArrays acquired data with a varying sample interval between 30-min to 1 h. The west lobe ShapeArray was not recording data from January 14 at 11:00 to January 15 at 13:00 for reasons unknown. To compare the ShapeArray data with the DAS data, we selected a common baseline reference to review relative displacement. We selected the closest available time (January 12 at 12:00) based on the ShapeArray sampling interval to the beginning of the DAS recording (January 12 at 11:38). The x-direction displacement at the shallowest sensor (at -0.3-m depth) was selected for comparison with DAS. A full-depth profile of the west lobe and east lobe ShapeArrays is included in Fig. S8, illustrating changes over the three-day period. Based on the lateral deformation along the ShapeArray installation depth, we observe an initiation of movement at 1.5 m depth at the west lobe ShapeArray, and at 1.0 m depth at the east lobe ShapeArray. Deformation is observed to be distributed towards the surface. We select two DAS channels at adjacent cable sections closely aligned with the ShapeArrays for comparison.

## Data availability

The DAS data generated in this study have been deposited in the Zenodo database under accession code https://doi.org/10.5281/zenodo.8356348.

## Code availability

The relevant code to reproduce the results of this research is available at: https://github.com/smouellet/hhdas.

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

## Acknowledgements

This research was funded by a Mitacs Accelerate International grant (IT31799, J.D.). The authors gratefully acknowledge the support of industry partners, BGC Engineering and OptaSense. We thank Andres Chavarria and Victor Yartsev of OptaSense for providing technical support and for sharing the collected DAS dataset jointly with the British Geological Survey. We thank Andreas Fichtner for comments on an earlier version of the work. The British Geological Survey shared the collocated instrumentation data and lidar scans. Ben Dashwood and Jonathan Chambers publish with the permission of the Executive Director, British Geological Survey (UKRI-NERC).

## Author contributions

Conceptualization: S.M.O., J.D., and D.J.H. Methodology: S.M.O. and J.D. Investigation: S.C., R.C., B.D., and J.E.C. Visualization: S.M.O. and M.J.L. Supervision: J.D. Writing—original draft: S.M.O. Writing—review and editing: S.M.O., J.D., M.L., M.K., D.J.H., R.C., S.C., B.D., and J.E.C.

## Competing interests

The authors declare the following competing interests: OptaSense Ltd. (owned by Luna Innovations) is a funder of this research, via a Mitacs Accelerate International grant. At the time of producing this paper Roger

Crickmore and Steve Cole were employed by Optasense Ltd, although Steve has since retired. The production of such papers is considered to be part of their normal duties as employees of Optasense, and they will not gain financial advantage due to its publication. The interrogator used to make the measurements does include some patented techniques, but these are not discussed in this paper so we do not believe that the value of any of these patents will be affected by its publication. Having their instrumentation mentioned in a peer reviewed paper is obviously of some benefit to Optasense as it slightly raises the profile of the company. However, given the range of other activities (trade shows, conferences, webinars, etc) that Optasense also undertake, the benefit from the publication of a single paper will likely be very small and impossible to quantify. We feel that it is standard practice in scientific papers and conference presentations to mention the source of any specialist equipment that was used to make the measurements, especially when the developers of the equipment have assisted in the analysis of its data. The remaining authors declare no competing interests.
