## [Peer Review File · Nature Communications]

Previously hidden landslide processes revealed using distributed acoustic sensing with nanostrain-rate sensitivityREVIEWER COMMENTS

Reviewer #1 (Remarks to the Author):

The paper provides a comprehensive analysis of a rainfall-induced slow-moving landslide event using a DAS strain data at low frequency. The results are of great interest and offer extremely valuable insights into the dynamics of slow-moving landslides. In addition, as highlighted numerous times in the paper it brings promising results for the development of early warning systems. However, the paper might need some improvements before being considered for publication. Notably, to enhance the overall flow of the paper, it is advisable to eliminate repetitions and maintain consistency in terminology and units.

Here are specific suggestions to help improving the manuscript:

Introduction:

- L54: Consider rephrasing the term "survey".
- L92: Consider using consistent units for clarity and comparability.

Background:

- L142: Strain change is described but you only refer to strain and strain rate. It would be nice to include the units and maintain consistency throughout the text.
- Fig 1: Consider enhancing clarity by using a different method to highlight profile b without arrows (maybe circle it in black) and improving the readability of the LiDAR background.
- Consider removing reference to CIF formation in the map since it's not visible. In addition, in caption "Cleveland Ironstone Formation 130 (CIF) and Staithes Sandstone Formation (SSF) geological boundaries are per Merritt et al, 2014." There is no reference to the CIF formation in Merritt et al, 2014.
- Consider clarifying the meaning of arrow length in the map caption if there's a specific meaning to these.
- L181: "corresponds": would alternative terms like "correlates" or "is associated" be more appropriate?

You wrote: “We estimate an average retrogression velocity of ~1.7 m/hour, as the strain-rate front propagates over a 20-m distance in a 12-hour period.” I assume the retrogression velocity is computed as 20 divided by 12. Is the 20 meters indicative of continuous displacement over 12 hours, a deformation propagating along a 20-meter transect, or a 20-meter transect with movement during 12 hours, possibly peaking at 12mm/day?

In the latter cases, inferring a constant retrogression velocity of 1.7m/hr may not accurately represent the observed dynamics at Hollin Hill. Figure 2 suggests a combination of the two last scenarios, involving both transect propagation and intermittent movement within the 20-meter distance. May be interesting to clarify and be careful with the terminology you use to clarify the message.

L194: “2.5 m/hour; Fig. S1, Sequence 3B” consider adding “Fig2” as well.

I guess that the velocity of 2.5m/hr corresponds to the propagation of the displacement front and not the landslide velocity?

L195-197: Would it more accurate to the propagation velocity rather than the velocity itself in this context?

L201: You use microstrain (1000) and have figures with millistrain and microstrain.

Fig3B, You changed the scale of millistrain to microstrain, using -0.2 to 0.2 millistrain would have been fine.

To improve the clarity consider using the same unit everywhere.

L235-237: The reference provided, 'Whiteley, J. S., et al. 2019,' does not pertain to the use of satellite imagery.

The order of figure citation is not continuous, Fig 6 cited in the Fig 3 caption and appearing before Fig 5.

To enhance readability, it might be beneficial to draw the scarps on Figure 4, preventing the need to cite Fig 6 in L239 and maintaining a more logical order.

Regarding Fig 5, if the positions of the lobes shown by LiDAR are consistent, it's worth reconsidering the relevance of comparing the east SAA and DAS at the green dot. The green dot appears to be situated between two lobes, potentially indicating minimal displacement. The east SAA, on the other hand, is positioned at the border or within a lobe itself. While this comparison may hold value for assessing temporal changes, it can be interesting to include other lines in the comparison.

Discussion:

- Consider clarifying the meaning of the term "retrogression velocity" and its calculation. Maybe by reevaluating the interpretation of rapid movement in terms of "displacement front propagation velocity" (or something shorter) rather than actual velocity.
- L275: Consider using terms like "small variations" instead of "slow variances."
- Consider clarifying the relevance of citing Hungr and Varnes in the context of landslide hazard assessments.
- Consider limiting repetitions (The 2 first paragraphs end with the same conclusion).
- Consider include a conceptual drawing to replace fig6, illustrating the interpreted sliding processes more clearly. Also, Fig 6 is not cited in the discussion.

L358-361: this assumption seems a bit risky with only 2 SAA to compare with at places relatively close.

In overall, the discussion could be streamlined to eliminate repetition and directly address key points. Additionally, it would be beneficial to emphasize the limitation of only monitoring surface displacement and its impact on understanding the landslide's dynamics. Can the method effectively capture or infer deformation processes at depth? While the approach shows promise for depicting pattern of surface movements and providing early warnings, the accuracy of quantifying deformation is debatable, especially considering spatial heterogeneity and temporal variations during events in coupling. A constructive discussion on potential method improvements for a more comprehensive understanding would be valuable.

Material and Methods:

- Consider providing additional details about discrete points in L394-396, emphasizing their position relative to the gauge.
- Consider clarifying the definition of velocity, differentiating between L188 and L191.
- L440: consider adding a paragraph for "The remainder..."
- Consider clarifying the distinction between phase 3 and 4 and ensure consistency with terms like "relative strain." That you never mentioned before.
- L456: Are you suggesting that the fiber optic's location is determined through interpolation

of positions at 'nb' points, measured along the fiber at 20-meter intervals using an RTK GPS system (LEICA, ...)?

- Consider removing "complementary ..." L460-461.

- L523: unit missing

L524-526: you already mentioned it L504-505.

I noticed that all sections were named except the first one.

Perhaps it would be beneficial to move the 'Temperature Effect' section to the beginning and potentially streamline or reduce its content since it has been neglected.

Supplementary:

- Consider organizing supplementary materials in the order of their citation in the main paper.

Reviewer #2 (Remarks to the Author):

Dear authors,

Thank you for your very interesting contribution. Please find my comments and suggestions in the attached document.

Sincerely

[Editorial note: Please find reviewer #2's attached document beginning on the next page.]

Review of: “Previously hidden landslide processes using distributed acoustic sensing with nanostrain-rate sensitivity”, by Ouellet et al

This manuscript describes the use of low-frequency Distributed Acoustic Sensing (DAS) for monitoring the spatio-temporal evolution of landslide movements at the well-characterized Hollin Hill Landslide Observatory in the UK. The study develops a processing flow to extract reliable low-frequency (< 1 Hz) strain and strain-rate information from the DAS data which shows the spatial and temporal evolution of deformation within the landslide during a period of movement reactivation due to rainfall. Strain-rate and strain data are evaluated to determine 4 different deformation sequences affecting different zones of the landslide, which describe the landslide movement from reactivation of the scarp, to development of a rupture zone in the central area of the landslide and retrogression to the scarp and final stabilization. The low-frequency DAS measurements are also used to estimate displacements that are compared with independent deformation measurements from a ShapeArray, with good agreement. The paper clearly demonstrates that low-frequency DAS has potential as a monitoring tool in shallow landslides.

First, I would like to start by thanking the authors for this very interesting study. I find results exciting and very encouraging, and I think that this paper is an important contribution for the DAS and landslide monitoring community. The paper is in general well written and illustrations are of good quality, clear and illustrative of the work carried out. I also appreciate the authors considering effects such as temperature changes on the strain data, as this is one of the significant issues when interpreting low-frequency DAS measurements.

I think that this paper is valuable and should be published. However, I also think that some changes and additions could be made, which would improve the clarity and impact of the paper. Below, I have provided some general and some more detailed suggestions that I hope the authors find useful to improve this interesting contribution.

Major general comment

The Hollin Hill landslide has been monitored for a long time. A lot of work has been done and publications exist describing the inferred landslide structure and dynamics and discussing conceptual models for the landslide. However, not much of this work is mentioned in the paper. I encourage you to take advantage of this information available, and use it to do some more comparisons between what is learned and interpreted here from the DAS data, and what is already known about the landslide. The paper mentions that this investigation illuminates landslide processes that could not be resolved before, but it never really discusses what is already known and what exactly new information is being provided by the strain data. Some more background information and discussion on this respect is needed to clarify what are the new insights gained from this approach.

Detailed comments on Results

In general, I find this section a little bit difficult to follow. The observations are very complex (which is great, it means that you can see a lot), but then you need to be very careful and clear when you are describing the results. My general suggestion is that you always start by describing what you see in the data, and then continue to say what you interpret that observation to be. Also, I suggest that you be specific about whether you are referring to strain-rate or strain (which sometimes can show different trends depending on where you are in the temporal evolution curve). Another suggestion is to make specific references to Figures and the

location of the features that you are describing (specially when you refer to Figure 2). For the reader who is not familiar with these types of measurements and cannot remember the morphology of the landslide just by looking at Figure 1 once, it will be difficult to know exactly where to look in each of the panels if you don't mention it.

- I think it would be most helpful to already indicate in Figure 1 the location of the chosen channels described in this section and plotted in Figure 3. I think it would also be helpful to mark the position of these channels in Figure 2; that would make the comparison of Figures 2 and 3 easier.

- Line 174: "Sequences 1 through 4" - Please specify here that you are referring to the numbered squares in Figure 2.

- I think you don't discuss anything in the text, but there seems to be a relatively prominent anomaly developing in the lower half of all profiles (within the flow dominated area of the landslide) at the time of maximum soil moisture content (e.g. seen between channels 1140 and 1160 on L1). This anomaly seems to be further north as we move eastwards. Do you have any interpretation about what this feature could be based on what you know about the landslide?

- Minor comment: I suggest to add a mention to Figure S2 here (somewhere in/after lines 163/166), so that the reader can compare the strain-rate images in Fig. 2 with the strain panels in Fig. S2.

Strain initiation at head scarp

- I am a bit confused about your observed "onset of strain" in Sequence 1. In the text (lines 177-182) you describe a "steady increase in strain near the main scarp, with an averaged inferred velocity of 1 mm per day". Although slightly positive strain is observed in the upper few channels in Figure 2 for L6, the black line in Figure 3B shows decreasing and not increasing strain. The gray line seems to indicate positive strain-rate just before the start of Sequence 1 (just before the yellow band), but it is immediately followed by a decreasing strain-rate. Although this onset strain signal might be there, I don't think it is clearly observable in these Figures.

- Line 187: "... develops increasing strain centered around cable section L4" - A reference to the corresponding panel in Figure 2 would be helpful here.

Triggering of rupture zone (Sequence 2), retrogression towards scarp (Sequence 3A) and flow lobe surge (Sequence 3B)

- I also found the description of Sequences 2 and 3 difficult to follow. Two things that make this description confusing for me:

1) You describe strain evolution in terms of inferred velocities, but your Figures are all displayed in units of strain or strain-rate. It would make it easier to link the text to the Figures if you talk about strain/strain-rate values first, and then mention what this means for the landslide processes in terms of inferred velocities.

2) You talk about "strain" without mentioning if it is positive strain or negative strain. For example, in the description of Sequence 2 (lines 187-188), you mention "The strain accelerates to a maximum inferred velocity of 12 mm/day at 12:00 on January 14 (Fig. 3A, sequence 2)". When I look at the corresponding panel in Figure 3A, I see decreasing strain and strain-rate, and a large negative anomaly in L4 in Figure 2. So, in this case, you are referring to a negative strain/strain-rate anomaly, which indicates compression. It is important that you mention that during your description, otherwise it is difficult for the reader to follow, since they have to look at two figures and read at the same time. Describing the observations in terms of "extension" and "compression" will make it clearer to understand what the strain/strain-rate data shows.

- When you describe Sequence 3A, you start by saying “Here, a general retrogression of the rupture zone towards the scarp occurs”. This is an interpretation, not an observation. Thus, I suggest that you start by saying that you observe a relatively sharp increase in strain/strain-rate, which you interpret as a retrogression of the rupture zone.

- Lines 192-193: “ This coincides with accelerating strain near the main scarp” - Please add a reference to the panel in Figure 2 where you can see this, and mention channel ranges so that the reader can easily locate the area and feature you are referring to.

- Line 193: “ Further downslope...” - As above, where can I see this? Please refer to the Figures and the channel ranges I should be looking at.

Stabilization of rupture zone with gradual deceleration in strain at scarp (Sequence 4)

- Lines 201-205: “A maximum strain of ~1000 microstrain occurs at the rupture zone...” and “the strain continues to increase at more gradual levels at the main scarp...” - Please indicate where you can see these features in Figures 2 and 3.

Discussion

- Lines 297-308: I think this is a nice summary of the dynamics that you can see in the strain-rate data. Are these interpretations solely based on your data, or are they also informed by previous work done on this landslide? I think that your results agree nicely with the observations and conceptual models developed in Uhlemann et al (2016)*, Figures 1 and 10. Specially, Figure 10 of this paper shows their interpreted compression and tension zones in the shallow subsurface of the landslide, which I think agree reasonably well with the profile that you show in Figure 1b (which is very nice to see!). Although this could have changed since then due to the rapid movement of the landslide, I think it would be interesting to look into it and compare your findings with theirs. You can even consider reproducing the figure (with permission) and show it along side your profile if it makes sense.

*Uhlemann, S., Hagedorn, S., Dashwood, B., Maurer, H., Gunn, D., Dijkstra, T., & Chambers, J. (2016). *Landslide characterization using P- and S-wave seismic refraction tomography – The importance of elastic moduli*. *Journal of Applied Geophysics*, 134, 64–76. <https://doi.org/10.1016/j.jappgeo.2016.08.014>

- Lines 315-316: “Our observations are consistent with earlier studies that demonstrate the influence of seasonal precipitation on slope movement”. There is a lot of monitoring work done on this landslide using different types of geophysical techniques (e.g. ERT and seismic monitoring). However, here you only compare it with the ShapeArray data. This section should include a bit more (general) discussion on what these other geophysical techniques show that is relevant to your study, and how your results provide more/complementary information about the landslide processes. I don’t suggest a detailed comparison as what you have done with the ShapeArray data, but a description of what other works show and how they compare to your observations would be granted here.

- Lines 338-339: “Future research should investigate the effect of cable geometry on the detectable strain thresholds.”. I agree. Along the same lines, how about the effect of cable construction on the measured/detectable strain magnitudes? I know that you only had one type of cable, but can you (very briefly) comment on how using loose-tube, gel-filled cable, for example, would affect strain measurements? A brief mention is sufficient since it is beyond the scope of your paper to compare cables, but I think it is an important

consideration when planning deployments for low-frequency DAS strain monitoring and it is worth mentioning. You could also discuss this after the discussion on strain transfer in comparison with ShapeArray (ending in line 361).

- Throughout the discussion, I think it is worth making clear that this is a shallow landslide and movements are in the near-surface, where the cable is deployed. The methods could be less effective for deep-seated landslides where the movements happen at depth.

Materials and Methods

- How long are the data files recorded by the IU? Do you concatenate the data for the entire period of interest before you conduct your decimation, or do you apply your decimation file by file? Are steps 6 and 7 applied to individual files or a long file encompassing the entire recording period? Please clarify.

- Line 458-460: " Short period of data gaps..." - DAS measures relative strain change, and stopping and restarting the measurements causes losing your reference strain state, sometimes. Can you comment on how large these data gaps are and how you deal with them in your integration? How do you make sure that you are not seeing a change in strain at each data gap?

- Line 485: ")" needs to be removed.

- I appreciate the effort to calculate the effect that temperature changes affecting the cable can have on strain measurements. Did you observe any effects associated with temperature changes affecting the IU itself? They should manifest as "common-mode noise" along the cable. Did you have any temperature control in the barn?

- Another way of monitoring temperature changes along the fiber would be to deploy DTS along the same cable (different fiber). I suggest mentioning this in the section describing the temperature effect evaluation, or even in the Discussion section of the paper, as an additional consideration for future experiments of this type.

General Minor suggestions

- I suggest mentioning the name of the study site already in the Abstract. It is mentioned for the first time in the Introduction with not much context (lines 78-79), and then directly introduced in the Background section.

- Why is the photo of fiber installation the very last one in the Supplementary (Fig. S6) if it is the first one to be mentioned in the text? I would suggest moving it to be the first one.

- Line 146: ".. n corresponds to a refractive index" - I would be a bit more specific and say "refractive index of the fiber"

- Line 409-410 : " Selecting a channel spacing less than the LG is akin to applying a moving average filter to the dataset" - I think this sentence is a bit confusing and could be rephrased. The gauge length is what acts as a moving average filter. So, I would rather say that when channel spacing is smaller than the gauge length, the effect of the gauge length is the same as applying a moving average to the data, moving one channel at a time (or something along these lines).

- Figure 3: the time labels seem to be displaced with respect to the tick marks, which makes it a bit confusing.

- Figure 5: is there a data gap In the ShapeArray data on the West lobe? I think it is not mentioned in the text.

- Figure 5: it might be useful to plot precipitation and soil moisture on top of the displacements, for completeness and aid reference to the previous figures and observations.

Reply to peer reviewer comments

Comments from Reviewer 1:

The paper provides a comprehensive analysis of a rainfall-induced slow-moving landslide event using a DAS strain data at low frequency. The results are of great interest and offer extremely valuable insights into the dynamics of slow-moving landslides. In addition, as highlighted numerous times in the paper it brings promising results for the development of early warning systems. However, the paper might need some improvements before being considered for publication. Notably, to enhance the overall flow of the paper, it is advisable to eliminate repetitions and maintain consistency in terminology and units.

Response: Thank you for providing your valuable input and constructive feedback on our manuscript. We have addressed your feedback to improve the manuscript, including eliminating any repetitive statements and ensuring consistency with terminology and units. Our point-by-point responses are below.

Here are specific suggestions to help improving the manuscript:

Introduction:

- L54: Consider rephrasing the term "survey".

Response: Thank you for your suggestion. We have rephrased this term for clarity.

Applied changes:

- L53: Rephrased "survey" as "geodetic surveying"

- L92: Consider using consistent units for clarity and comparability.

Response: We have revised our statement to use consistent units.

Applied changes:

- L90-92: *"For example, monitoring methods such as ground-based interferometry and automated inclinometers (i.e., ShapeArrays) can resolve up to sub-millimeter displacements over minute timescales, but here we resolve changes in the order of nanometers with 1 Hz sampling frequency"*

Background:

- L142: Strain change is described but then you only refer to strain and strain rate. It would be nice to include the units and maintain consistency throughout the text.

Response: Thank you for pointing out these important inconsistencies in our terminology and units. We have incorporated several sentences to clarify the units. revised our manuscript to confirm consistent usage of 'strain change' and have added in a clarification note below.

Applied changes:

- L164-168: *"The optical phase change and corresponding strain measurement represent a change from a baseline measurement occurring at the start of data acquisition. For clarity, we underscore that all references to strain ϵ and strain-rate $\dot{\epsilon}$ herein represent a relative measurement from our baseline measurement on January 12, 2021 at 11:38 UTC."*

- Fig 1: Consider enhancing clarity by using a different method to highlight profile b without arrows (maybe circle it in black) and improving the readability of the LiDAR background.

Response: Thank you for your suggestion. We have updated Figure 1 accordingly.

Applied changes:

- Figure 1: Included a black outline to highlight Profile B and remove annotation. Updated figure caption to highlight Profile B captured in black outline.

- Consider removing reference to CIF formation in the map since it's not visible. In addition, in caption "Cleveland Ironstone Formation 130 (CIF) and Staithes Sandstone Formation (SSF) geological boundaries are per Merritt et al, 2014." There is no reference to the CIF formation in Merritt et al, 2014.

Response: Thank you for pointing this out. We have removed the reference to the CIF formation on the map and from the caption.

Applied changes:

- Figure 1: Removed reference to CIF from map and caption.

- Consider clarifying the meaning of arrow length in the map caption if there's a specific meaning to these.

Response: Thank you for your suggestion. The length of the arrows are qualitative. We've revised our caption to clarify this more appropriately.

Applied changes:

- Figure 1. Revised caption as follows: "*Strain vector lengths are qualitative and do not represent actual strain magnitudes.*"

- L181: "corresponds": would alternative terms like "correlates" or "is associated" be more appropriate?

Response: We agree with your suggestion and feel that "correlates" would be more appropriate. We've incorporated this change into our revision.

Applied changes:

- L243-244: "*The change in strain correlates with a steady increase in soil moisture content from 43 to 47%.*"

You wrote: "We estimate an average retrogression velocity of ~1.7 m/hour, as the strain-rate front propagates over a 20-m distance in a 12-hour period." I assume the retrogression velocity is computed as 20 divided by 12. Is the 20 meters indicative of continuous displacement over 12 hours, a deformation propagating along a 20-meter transect, or a 20-meter transect with movement during 12 hours, possibly peaking at 12mm/day? In the latter cases, inferring a constant retrogression velocity of 1.7m/hr may not accurately represent the observed dynamics at Hollin Hill.

Response: Correct, we computed the retrogression velocity as 20 divided by 12. The 20 meters is indicative of a deformation propagating along a 20-meter transect. In consideration of your comments, we have chosen to remove a reference to this retrogression velocity to improve the clarity of this section. We have reworded this section as per our changes below.

Applied changes:

- L254-258: We modified this statement and removed the reference to the average retrogression velocity to improve clarity. The revised statement is as follows: "*From the triggering of the rupture zone (January 14 at ~3:00) to ~12 hours later, the strain-rate front propagates from the rupture zone upslope towards the scarp (over a ~20 m distance; Fig. 2B). We interpret this as a retrogressive deformation of the landslide (Sequence 3A). This coincides with accelerating strain near the main scarp (Fig. 4D).*"

Figure 2 suggests a combination of the two last scenarios, involving both transect propagation and intermittent movement within the 20-meter distance. May be interesting to clarify and be careful with the terminology you use to clarify the message.

Response: Thank you for your comments. We have made significant changes to this section which we feel help the improve clarity of our interpretation.

Applied changes:

- Created new figure to support interpretation (**Figure 3. Conceptual framework to support interpretation of strain-rate patterns**).
- L202-217: Added additional explanation of the strain-rate observations and their interpretation.

L194: “2.5 m/hour; Fig. S1, Sequence 3B” consider adding “Fig2” as well. I guess that the velocity of 2.5m/hr corresponds to the propagation of the displacement front and not the landslide velocity?

Response: Correct, the velocity of 2.5 m/hr corresponds to the propagation of the strain (displacement) front. We describe this in our manuscript as "propagation of the strain front at the flow lobe". Thank you for your suggestion, we agree that it is appropriate to add Figure 2 here as well. We have also revised Figure 2 to improve the clarity and aid referencing the corresponding event sequences.

Applied changes:

- Figure 2. Revised for clarity (only including select cable section referenced within the text, highlighting the corresponding event sequences appropriately within the figure).
- L262: Updated figure references: "*~2.5 m/hour; Fig. 2D, Fig. S4*"

L195-197: Would it more accurate to the propagation velocity rather than the velocity itself in this context?

Response: Per above. We have revised our wording of the velocity for clarity.

Applied changes:

- L260-265: "*Further downslope, propagation of the strain-rate front at the flow lobe occurs across ~10 m in the downslope direction over a four-hour period (average velocity of the strain-rate front propagation is ~2.5 m/hour; Fig. 2D, Fig. S4). This is interpreted as a surge of superficial material flowing over the cable (not representative of the motion of the rupture zone assessed above, where the material is assumed to be well coupled with the cable, see Discussion).*"

L201: You use microstrain (1000) and have figures with millistrain and microstrain.

Fig3B, You changed the scale of millistrain to microstrain, using -0.2 to 0.2 millistrain would have been fine. To improve the clarity consider using the same unit everywhere.

Response: Thank you for your suggestion. We have chosen to only use units of millistrain when describing and displaying strain measurements for clarity and consistency.

Applied changes:

- L271: Upon revision, we have removed this statement ("*A maximum strain of...* ") as we feel it is not necessary for the explanation of the event sequence and decreases clarity of the text.
- L544: Revised from microstrain to millistrain "*As such, a change in 1 °C corresponds to change of $1.1 \cdot 10^{-2}$ millistrain*"
- Figure 4F (former Figure 3B) Revised units to millistrain.

L235-237: The reference provided, 'Whiteley, J. S., et al. 2019,' does not pertain to the use of satellite imagery.

Response: Thank you for pointing out this error. We have updated this with the appropriate reference 'Whiteley, J.S. et al. 2020. Landslide monitoring using seismic refraction tomography - The importance of incorporating topographic variations'. We also revised our sentence for accuracy.

Applied changes:

- L303: Replaced "satellite imagery" with "aerial photography" and updated the corresponding reference.

The order of figure citation is not continuous, Fig 6 cited in the Fig 3 caption and appearing before Fig 5.

Response: Thank you for your feedback. We have revised all figure citations to be in continuous order of citation.

To enhance readability, it might be beneficial to draw the scarps on Figure 4, preventing the need to cite Fig 6 in L239 and maintaining a more logical order.

Response: Thank you for your suggestion. We agree with this approach and have updated Figure 4 (now Figure 5) accordingly.

Applied changes:

- Updated Figure 4 (now Figure 5) to include scarps. We have also removed the former Figure 6 as we felt it decreased overall clarity of the manuscript.

Regarding Fig 5, if the positions of the lobes shown by LiDAR are consistent, it's worth reconsidering the relevance of comparing the east SAA and DAS at the green dot. The green dot appears to be situated between two lobes, potentially indicating minimal displacement. The east SAA, on the other hand, is positioned at the border or within a lobe itself. While this comparison may hold value for assessing temporal changes, it can be interesting to include other lines in the comparison.

Response: Thank you for your suggestions. We have revised our figure comparing the SAA and DAS channels to include two additional DAS channels in the comparison.

Applied changes:

- Revised Figure 5 (now Figure 6) to include DAS channels from L3 and L4. We have also added in the precipitation and moisture content to this figure, per a separate reviewer suggestion.

Discussion:

- Consider clarifying the meaning of the term "retrogression velocity" and its calculation. Maybe by reevaluating the interpretation of rapid movement in terms of "displacement front propagation velocity" (or something shorter) rather than actual velocity.

Response: Thank you for your comments. We have made significant changes to this section to clarify. Notably, we have created a new figure to support our interpretation of strain-rate patterns, and revised our language for clarity.

Applied changes:

- Created new figure (Figure 3) to support interpretation of results and clarify the meaning of terms such as retrogression velocity.
- Revised language to describe observations in Results.
- L354-357: *'Our study provides observations of strain-rate front propagation in the flow-dominant zone to characterize the spatial extents of an interpreted flow surge event occurring over an ~8-hour period, along with its associated strain-rate front propagation velocity of 2.5 m/hour.'*

- L275: Consider using terms like "small variations" instead of "slow variances."

Response: Thank you for your suggestion. Upon further review, we have revised this statement for improved clarity and conciseness.

Applied changes:

- L357: Reworded former statement of, *"Our method can infer small variations ... "* as follows: *"Our method can be used to distinguish kinematic zones ... "*

- Consider clarifying the relevance of citing Hungr and Varnes in the context of landslide hazard assessments.

Response: Thank you for your feedback. We have revised this section to improve clarity of our citation by adding a separate statement with the Hungr et al. (2014) citation.

Applied changes:

- L357: Removed Hungr et al. (2014) citation from the following sentence: "*Our method can be used to distinguish kinematic zones which vary over time to support landslide hazard assessment ...*"
- L361-363: Added the following statement with the Hungr et al. (2014) citation: "*... improves our ability to characterize landslide events and behaviours using current classification systems.*"

- Consider limiting repetitions (The 2 first paragraphs end with the same conclusion).

Response: Thank you for your helpful input. We have revised this section accordingly.

Applied changes:

- L365-367: Modified the conclusion of the initial paragraph by merging with information from following paragraph, as follows: "*Therefore, our DAS-based method can provide crucial information for landslide early-warning applications by enabling monitoring over broad temporal and spatial scales.*"
- L367: Deleted the following sentence from the end of the second paragraph to avoid repetition: "*For landslide early-warning applications, DAS strain rates (and the inferred velocities) could become an important tool to enable monitoring over broad temporal and spatial scales.*"

- Consider include a conceptual drawing to replace fig6, illustrating the interpreted sliding processes more clearly. Also, Fig 6 is not cited in the discussion.

Response: Thank you for your suggestion. We have included a separate drawing to support our interpreted sliding processes (new Figure 3) and have removed the former Figure 6 for clarity.

Applied changes:

- Figure 3 (created new figure, titled: *Conceptual framework to support interpretation of strain-rate patterns.*).

L358-361: this assumption seems a bit risky with only 2 SAA to compare with at places relatively close.

Response: Thank you for your comment. We have added an additional statement here to address this.

Applied changes:

- L440-443: "*However, with only two ShapeArrays sensors for comparison, and one of them at a location offset from the flow lobe, further studies on the inferred quantitative DAS-derived displacements are needed to provide a more robust estimate of these assumptions.*"

In overall, the discussion could be streamlined to eliminate repetition and directly address key points. Additionally, it would be beneficial to emphasize the limitation of only monitoring surface displacement and its impact on understanding the landslide's dynamics. Can the method effectively capture or infer deformation processes at depth? While the approach shows promise for depicting pattern of surface movements and providing early warnings, the accuracy of quantifying deformation is debatable, especially considering spatial heterogeneity and temporal variations during events in coupling. A constructive discussion on potential method improvements for a more comprehensive understanding would be valuable.

Response: Thank you for your suggestions. We have made several changes to our discussion to address your comments.

Applied changes:

- L414-418: *"Future research should investigate the effect of cable geometry on the detectable strain thresholds and on the accuracy of quantifying deformation. For example, acquiring DAS data from a downhole fiber optic cable paired with an inclinometer or ShapeArray, combined with a trench installation would provide useful information on the capabilities of DAS to capture deformations occurring at depth."*
- L440-443: *"However, with only two ShapeArrays sensors for comparison, and one of them at a location offset from the flow lobe, further studies on the inferred quantitative DAS-derived displacements are needed to provide a more robust estimate of these assumptions."*

Material and Methods:

- Consider providing additional details about discrete points in L394-396, emphasizing their position relative to the gauge.

Response: Thank you for your helpful input and constructive feedback on our manuscript. We have included additional details accordingly.

Applied changes:

- L486-489: *"The DAS measurement corresponds to a discrete point (i.e. channel) along the fiber, representing an average measurement which is centered over a length of fiber referred to as the gauge length (L_G)^{51,57} (Fig. 7)."*

- Consider clarifying the definition of velocity, differentiating between L188 and L191.

Response: Thank you for your suggestion. We have made substantial changes to this section as per above, including the creation of a new figure that we feel helps the reader to follow our interpretation of the results. We have made changes to this section as follows.

Applied changes:

- L204-207: *"Due to the spatiotemporal characteristics of the DAS dataset, velocity features of the landslide are obtained per the following approaches: (1) the velocity occurring at a single location (i.e. DAS channel), (2) the velocity of a strain or strain-rate front propagating over multiple DAS channels with time."*
- L246-252: *"On January 14 at ~3:00, a slip surface pattern develops ~30 m southwest of the main scarp at cable sections L3 and L4 (Fig. 2). Based on the greater magnitude of the observed strain changes at this location and over this time period, in comparison with other cable locations, we interpret this as a triggering of the rupture zone. The strain rate decreases to a minimum value of -35 nm/m*s (inferred velocity of -12 mm/day) at 12:00 on January 14 at cable section L4 (Fig. 3A and Fig. 2, Sequence 2). Over this period, the soil moisture content increases from 47% to a maximum of 53%."*

- L440: consider adding a paragraph for "The remainder..."

Response: Thank you for your suggestion. We agree that this will help clarify this section and have revised accordingly.

Applied changes:

- L569: Removed numbered bullet and moved into standalone paragraph.
- L569-574: Added a paragraph to clearly separate processing steps using open-source Python modules and initial data acquisition using proprietary software.

- Consider clarifying the distinction between phase 3 and 4 and ensure consistency with terms like "relative strain." That you never mentioned before.

Response: Thank you for this important point. We have revised our manuscript to include details on the definitions of the changes in strain representing a relative measurement (L169-171)

Applied changes:

- L511: Removed "relative" from sentence for consistency with manuscript nomenclature and earlier definition.
- L166-168: *"All references to strain ϵ and strain-rate $\dot{\epsilon}$ herein represent a relative measurement from the baseline measurement on January 12, 2021 at 11:38 UTC."*

- L456: Are you suggesting that the fiber optic's location is determined through interpolation of positions at 'nb' points, measured along the fiber at 20-meter intervals using an RTK GPS system (LEICA, ...)?

Response: Correct.

Applied changes: N/A

- Consider removing "complementary ..." L460-461.

Response: Thank you for your suggestion. We have removed "complementary" from this sentence.

Applied changes:

- L601: ~~"Complementary-Geotechnical site data used to support our interpretation ..."~~

- L523: unit missing

Response: Thank you for pointing this out. We have added in the unit.

Applied changes:

- L544: *"As such, a change in 1 °C corresponds to change of $1.1 \cdot 10^{-2}$ millistrain."*

L524-526: you already mentioned it L504-505.

Response: Thank you for pointing this out. We have removed this sentence.

Applied changes:

- L548: Deleted the following repetitive sentence: *"We used a temperature sensor (installed at the weather station (Fig. 1A) at 10 cm depth) to estimate the relative contribution of temperature changes on the optical phase data."*

I noticed that all sections were named except the first one.

Perhaps it would be beneficial to move the 'Temperature Effect' section to the beginning and potentially streamline or reduce its content since it has been neglected.

Response: Thank you for suggestions. We added names to the initial sections and moved the "Temperature effects" section after the "Technology background", near the beginning of "Materials and Methods". We choose to maintain the content in the temperature section considering the Reviewer 2 comments.

Applied changes:

- L479: Added section name, "Technology background"
- L519: Moved "Temperature effects" to section following "Technology background"
- L553: Added section name, "Data processing"

Supplementary:

- Consider organizing supplementary materials in the order of their citation in the main paper.

Response: Thank you for your suggestion. We have revised the organization of the supplementary materials to follow the order of citation in the main paper.

Comments from Reviewer 2:

This manuscript describes the use of low-frequency Distributed Acoustic Sensing (DAS) for monitoring the spatio-temporal evolution of landslide movements at the well-characterized Hollin Hill Landslide Observatory in the UK. The study develops a processing flow to extract reliable low-frequency (< 1 Hz) strain and strain-rate information from the DAS data which shows the spatial and temporal evolution of deformation within the landslide during a period of movement reactivation due to rainfall. Strain-rate and strain data are evaluated to determine 4 different deformation sequences affecting different zones of the landslide, which describe the landslide movement from reactivation of the scarp to development of a rupture zone in the central area of the landslide and retrogression to the scarp and final stabilization. The low-frequency DAS measurements are also used to estimate displacements that are compared with independent deformation measurements from a ShapeArray, with good agreement. The paper clearly demonstrates that low-frequency DAS has potential as a monitoring tool in shallow landslides.

First, I would like to start by thanking the authors for this very interesting study. I find results exciting and very encouraging, and I think that this paper is an important contribution for the DAS and landslide monitoring community. The paper is in general well written and illustrations are of good quality, clear and illustrative of the work carried out. I also appreciate the authors considering effects such as temperature changes on the strain data, as this is one of the significant issues when interpreting low-frequency DAS measurements.

I think that this paper is valuable and should be published. However, I also think that some changes and additions could be made, which would improve the clarity and impact of the paper. Below, I have provided some general and some more detailed suggestions that I hope the authors find useful to improve this interesting contribution.

Response: Thank you for your helpful and constructive feedback. We have addressed your comments with details of our applied changes below.

Major general comment

The Hollin Hill landslide has been monitored for a long time. A lot of work has been done and publications exist describing the inferred landslide structure and dynamics and discussing conceptual models for the landslide. However, not much of this work is mentioned in the paper. I encourage you to take advantage of this information available, and use it to do some more comparisons between what is learned and interpreted here from the DAS data, and what is already known about the landslide. The paper mentions that this investigation illuminates landslide processes that could not be resolved before, but it never really discusses what is already known and what exactly new information is being provided by the strain data. Some more background information and discussion on this respect is needed to clarify what are the new insights gained from this approach.

Response: Thank you for your suggestion. We have incorporated additional information into our text to more clearly describe what is already known about the landslide and how our findings reveal new processes.

Applied changes:

Background:

- L107-118: "*Landslide processes at Hollin Hill have been studied using a variety of characterization and monitoring techniques, including seismic refraction tomography, geoelectrical resistivity, self potential, inclinometers, piezometers, lidar change detection, interferometric synthetic aperture radar (inSAR) and cone penetration testing*^{9,32,34-38}. Earlier studies have integrated multiple techniques to reveal different factors impacting landslide movement. For example, 3-D time lapse imaging of inferred slope moisture content (from inverted resistivity models) demonstrate moisture accumulation in the

upper slope over wintertime correlating with landslide reactivation and drainage from the mudstone to sandstone formation in the summertime³⁴. Seismic refraction tomography was then carried out along earlier geoelectrical resistivity profiles. The resulting ground model, incorporating both seismic and earlier resistivity modelling, describes zones of materials with different relative densities, leading to accumulating strains downslope. Drainage, occurring near the base of the flow lobes, results in stabilization of material near the toe³⁹."

Discussion:

- L345-363: *"We observe landslide processes with previously unresolved spatiotemporal resolution and demonstrate how these processes mimic those occurring over longer (i.e., seasonal) time scales. Our results are supported by findings from earlier studies describing similar movement patterns and that demonstrate the influence of seasonal precipitation and soil moisture content on slope movement^{34,35,38,39}. This agreement with earlier studies supports the validity of our findings. Our study reveals new insights into landslide dynamics, as it demonstrates the scale invariance of the earlier observed landslide processes because of the extreme sensitivity of our DAS-based method. For example, a past study hypothesized that deforming materials of different relative densities may contribute to a deformation wave progressing through the slope³⁹ (i.e., a flow surge event). Our study provides observations of strain-rate front propagation in the flow-dominant zone to characterize the spatial extents of an interpreted flow surge event occurring over an ~8-hour period, along with its associated strain-rate front propagation velocity of 2.5 m/hour (Fig. S4). Our method can be used to distinguish kinematic zones which vary over time to support landslide hazard assessments (e.g., discerning the motion characteristics of slower soil-creep events from more rapid flow-surge events). This new capability enables landslide monitoring on spatiotemporal scales that are currently poorly understood and improves our ability to characterize landslide events and behaviours using current classification systems⁴³."*

Detailed comments on Results

In general, I find this section a little bit difficult to follow. The observations are very complex (which is great, it means that you can see a lot), but then you need to be very careful and clear when you are describing the results. My general suggestion is that you always start by describing what you see in the data, and then continue to say what you interpret that observation to be. Also, I suggest that you be specific about whether you are referring to strain-rate or strain (which sometimes can show different trends depending on where you are in the temporal evolution curve). Another suggestion is to make specific references to Figures and the location of the features that you are describing (specially when you refer to Figure 2). For the reader who is not familiar with these types of measurements and cannot remember the morphology of the landslide just by looking at Figure 1 once, it will be difficult to know exactly where to look in each of the panels if you don't mention it.

Response: We appreciate your constructive comments. Upon further review, we agree that the observations are very complex, and the results may have been difficult to follow. To improve this, we have made significant changes to this section. Notably, we have added a conceptual framework (and corresponding figure) that outline how we obtain our interpretations from the strain-rate spatiotemporal patterns. Further details of our applied changes are below.

- I think it would be most helpful to already indicate in Figure 1 the location of the chosen channels described in this section and plotted in Figure 3. I think it would also be helpful to mark the position of these channels in Figure 2; that would make the comparison of Figures 2 and 3 easier.

Response: Thank you for your suggestion. We have updated Figures 1 and 2 to mark the position of these channels for clarity.

Applied changes:

- Figure 1: Added channel location labels on Figure 1A for comparison with Figure 3.
- Figure 2 (now 3): Added channel locations on Figure 2 for comparison with Figure 3.

- Line 174: “Sequences 1 through 4” - Please specify here that you are referring to the numbered squares in Figure 2.

Response: Thank you for your suggestion. We have modified this statement accordingly.

Applied changes:

- L229-234: *"Following our conceptual framework of strain-rate patterns (Fig. 3), we interpret the five event sequences which are ordered chronologically: (1) initiation of strain at the head scarp, (2) subsequent triggering of a rupture zone, (3) retrogressive strain towards the scarp, (4) a flow-lobe surge near the toe, and (5) stabilization of the rupture zone with a gradual increase in strain at the scarp. Sequences 1 through 5 are numbered and highlighted by dashed rectangles in Figs. 2b-e."*

- I think you don't discuss anything in the text, but there seems to be a relatively prominent anomaly developing in the lower half of all profiles (within the flow dominated area of the landslide) at the time of maximum soil moisture content (e.g. seen between channels 1140 and 1160 on L1). This anomaly seems to be further north as we move eastwards. Do you have any interpretation about what this feature could be based on what you know about the landslide?

Response: Thank you for highlighting this anomaly. The anomaly appears to be related to topographical effects of the landslide, as it occurs at a change of slope. We have revised our Results to include additional detail on general interpretation and include further details of this anomaly in our results and discussion. We have also included an additional figure in our Supplementary materials (**Fig. S4**) to help demonstrate the impact of topography on the resulting strain and strain-rate observations.

Applied changes:

- L216-217: *A sequence of paired extensional and compressional strain-rate observations are attributed to topographic and material strength variations (Fig. 3d).*
- L387-390: *"Additional observed strain-rate patterns are likely impacted by a combination of factors, including slope topography, strain redistribution from the shallow landslide movements, and variations in material parameters."*
- Supplementary Figure S4

- Minor comment: I suggest to add a mention to Figure S2 here (somewhere in/after lines 163/166), so that the reader can compare the strain-rate images in Fig. 2 with the strain panels in Fig. S2.

Response: Thank you for your suggestion. We have modified this statement accordingly. We have also modified our Figure 2 for clarity by only including strain-rate spatiotemporal images which are referenced in the main text, with figures of all six cable sections moved to the Supplementary (new Fig. S2).

Applied changes:

- L190-191: *"Strain spatiotemporal images are also available for comparison (Fig. S3)."*

Strain initiation at head scarp

- I am a bit confused about your observed “onset of strain” in Sequence 1. In the text (lines 177-182) you describe a “steady increase in strain near the main scarp, with an averaged inferred velocity of 1 mm per

day”. Although slightly positive strain is observed in the upper few channels in Figure 2 for L6, the black line in Figure 3B shows decreasing and not increasing strain. The gray line seems to indicate positive strain-rate just before the start of Sequence 1 (just before the yellow band), but it is immediately followed by a decreasing strain-rate. Although this onset strain signal might be there, I don’t think it is clearly observable in these Figures.

Response: Thank you for your comments. We have made significant revisions to this section to the clarity of our results and interpretations. We hope these changes increase the clarity of this section.

Applied changes:

- L202-217: Added an additional paragraph to illustrate our conceptual framework used to guide our interpretation.
- Figure 3. Created a new figure to support our interpretation.
- L238-244. Reworded this section for clarity, as follows: *"After about eight hours of rainfall, the onset of strain is observed at the northeast corner of the cable at the scarp (Fig. 4B, Fig. 4F). Over the subsequent eight-hour period, we observe a strain rate of $\sim -4 \text{ nm m}^{-1} \text{ s}^{-1}$ near the main scarp (corresponding to an average inferred velocity of -1 mm per day). The observed pattern at the scarp over this time corresponds to a low amplitude slip surface pattern (Fig 2E). The strain change correlates with a steady increase in soil moisture content from 43 to 47%."*

- Line 187: “... develops increasing strain centered around cable section L4” - A reference to the corresponding panel in Figure 2 would be helpful here.

Response: Thank you for your suggestion. We have made the following changes to increase clarity. Notably, we have revised our Figure 2 (strain-rate spatiotemporal images) to remove references to cable sections, instead adding lettering for clarity and only including segments referenced in the text.

Applied changes:

- L229-234: *"Following our conceptual framework of strain-rate patterns (Fig. 3), we interpret the five event sequences which are ordered chronologically: (1) initiation of strain at the head scarp, (2) subsequent triggering of a rupture zone, (3) retrogressive strain towards the scarp, (4) a flow-lobe surge near the toe, and (5) stabilization of the rupture zone with a gradual increase in strain at the scarp. Sequences 1 through 5 are numbered and highlighted by dashed rectangles in Figs. 2b-e.*
- L246-247: *"On January 14 at $\sim 3:00$ UTC, a slip surface pattern develops $\sim 30 \text{ m}$ southwest of the main scarp (Fig. 2c).*

Triggering of rupture zone (Sequence 2), retrogression towards scarp (Sequence 3A) and flow lobe surge (Sequence 3B)

- I also found the description of Sequences 2 and 3 difficult to follow. Two things that make this description confusing for me:

- 1) You describe strain evolution in terms of inferred velocities, but your Figures are all displayed in units of strain or strain-rate. It would make it easier to link the text to the Figures if you talk about strain/strain-rate values first, and then mention what this means for the landslide processes in terms of inferred velocities.

Response: Thank you for your suggestion. We believe we have addressed this comment as per our response to the following comment below.

Applied changes:

- See below.

- 2) You talk about “strain” without mentioning if it is positive strain or negative strain. For example, in the description of Sequence 2 (lines 187-188), you mention “The strain accelerates to a maximum inferred velocity of 12 mm/day at 12:00 on January 14 (Fig. 3A, sequence 2)”. When I look at the corresponding panel in Figure 3A, I see decreasing strain and strain-rate, and a large negative anomaly in L4 in Figure 2. So, in this case, you are referring to a negative strain/strain-rate anomaly, which indicates compression. It is important that you mention that during your description, otherwise it is difficult for the reader to follow, since they have to look at two figures and read at the same time. Describing the observations in terms of “extension” and “compression” will make it clearer to understand what the strain/strain-rate data shows.

Response: Thank you for your suggestion. We have made significant changes to this section to increase clarity. Notably, we have created a new Figure (Figure 3) that provides the primary strain-rate patterns which support our interpretation of rotational slip surface, retrogression, flow-lobe surge, topographic effects. We feel that this figure makes a significant improvement to aid the reader's understanding and avoids the requirement of adding additional text to describe the strain-rate observations.

Applied changes:

- Created new figure (Figure 3) to support interpretation of results, and revise Figure 2 for clarity.
- We have revised our text to use extension / compression descriptors and describe our interpretation as follows:
- L203-217: *"Positive and negative strain and strain-rate observations correspond to cable extension and compression, respectively, and we use the extensional or compressional descriptors herein. Due to the spatiotemporal characteristics of the DAS data, velocity features of the landslide are obtained per the following approaches: (1) the velocity occurring at a single location (i.e. DAS channel), (2) the velocity of a strain or strain-rate front propagating over multiple DAS channels with time. We guide our analysis using a conceptual framework to interpret key strain and strain-rate patterns, as follows: (1) Extensional strain-rate observations that are upslope of compressional strain-rate occurring over the same time period are interpreted as a shallow slip surface intersecting with the cable (Fig. 3a). (2) Propagation of the extensional strain-rate processes upslope with time are interpreted as slope retrogression. We characterize the retrogressive behaviour by analyzing the slope of the strain and strain-rate fronts with time (Fig. 3b). (3) A paired extensional and compressional strain-rate observation propagating downslope is interpreted as a surge of saturated materials (i.e., a flow surge) propagating over the cable (Fig. 3c). (4) A sequence of paired extensional and compressional strain-rate observations are attributed to topographic and material strength variations (Fig. 3d)."*
- Revised language to describe observations in Results.

- When you describe Sequence 3A, you start by saying “Here, a general retrogression of the rupture zone towards the scarp occurs”. This is an interpretation, not an observation. Thus, I suggest that you start by saying that you observe a relatively sharp increase in strain/strain-rate, which you interpret as a retrogression of the rupture zone.

Response: Thank you for your suggestion. See our above comments for changes. We have also revised our Sequence 3A description as per below.

Applied changes:

- L254-258: ***Retrogression towards scarp (sequence 3):** From the triggering of the rupture zone (January 14 at ~3:00) to ~12 hours later, the strain-rate front propagates from the rupture zone upslope towards the scarp (over a ~20 m distance; Fig. 2B). We interpret this as a retrogressive deformation of the landslide. This coincides with accelerating strain near the main scarp (Fig 4D).*

- Lines 192-193: "This coincides with accelerating strain near the main scarp" - Please add a reference to the panel in Figure 2 where you can see this, and mention channel ranges so that the reader can easily locate the area and feature you are referring to

Response: Thank you for your suggestion. We have modified this statement accordingly.

Applied changes:

- L257-258: "*This coincides with accelerating strain near the main scarp (Fig 4D).*"

- Line 193: "Further downslope..." - As above, where can I see this? Please refer to the Figures and the channel ranges I should be looking at.

Response: We have modified this statement and revised Figure 2 to improve clarity.

Applied changes:

- L233-238: "*Following our conceptual framework to interpret strain-rate patterns, we identify the following event sequences, demonstrating: (1) initiation of strain at the head scarp, (2) subsequent triggering of a rupture zone, (3) retrogressive strain towards the scarp and flow lobe surge, and (4) stabilization of the rupture zone with a gradual increase in strain at the scarp (Sequences 1 through 4, highlighted as numbered squares and circles in Fig. 2A and Fig. 2B, respectively).*"
- L260-262: "*Further downslope, propagation of the strain-rate front at the flow lobe occurs across ~10 m in the downslope direction over a four-hour period (average velocity of the strain-rate front propagation is ~2.5 m/hour; Fig. 2D, Fig. S4).*"

Stabilization of rupture zone with gradual deceleration in strain at scarp (Sequence 4)

- Lines 201-205: "A maximum strain of ~1000 microstrain occurs at the rupture zone..." and "the strain continues to increase at more gradual levels at the main scarp..." - Please indicate where you can see these features in Figures 2 and 3.

Response: We have revised this section for clarity.

Applied changes:

- L268-271: "*The final sequence coincides with a stabilization of the rupture zone, where the strain rate approaches near-zero values (Fig. 2D and Fig. 4E). The strain continues to increase at more gradual levels at the main scarp up until the end of the DAS acquisition period (Fig. 4A).*"

Discussion

- Lines 297-308: I think this is a nice summary of the dynamics that you can see in the strain-rate data. Are these interpretations solely based on your data, or are they also informed by previous work done on this landslide?

Response: Thank you for your feedback. These interpretations are based solely on our data. We have also incorporated an additional statement to this section as per the below.

Applied changes:

- L387-390: "*Additional observed strain-rate patterns are likely impacted by a combination of factors, including slope topography, strain redistribution from the shallow landslide movements, and variations in material parameters (Fig. 3D, Fig. SS)*".

I think that your results agree nicely with the observations and conceptual models developed in Uhlemann et al (2016)*, Figures 1 and 10. Specially, Figure 10 of this paper shows their interpreted compression and tension zones in the shallow subsurface of the landslide, which I think agree reasonably well with the profile that you show in Figure 1b (which is very nice to see!). Although this could have changed since then due to the rapid movement of the landslide, I think it would be interesting to look into it and compare your findings with theirs. You can even consider reproducing the figure (with permission) and show it along side your profile if it makes sense.

*Uhlemann, S., Hagedorn, S., Dashwood, B., Maurer, H., Gunn, D., Dijkstra, T., & Chambers, J. (2016). Landslide characterization using P- and S-wave seismic refraction tomography — The importance of elastic moduli. *Journal of Applied Geophysics*, 134, 64–76. <https://doi.org/10.1016/j.jappgeo.2016.08.014>

Response: Thank you for your suggestion. We have included an additional paragraph incorporating this.
Applied changes:

- L350-358: *"Our results are supported by findings from earlier studies describing similar movement patterns and that demonstrate the influence of seasonal precipitation and soil moisture content on slope movement^{34,35,38,39}. This agreement with earlier studies supports the validity of our findings. Our study reveals new insights into landslide dynamics, as it demonstrates the scale invariance of the earlier observed landslide processes because of the extreme sensitivity of our DAS-based method. For example, a past study hypothesized that deforming materials of different relative densities may contribute to a deformation wave progressing through the slope³⁹ (i.e., a flow surge event). Our study provides observations of strain-rate front propagation in the flow-dominant zone to characterize the spatial extents of an interpreted flow surge event occurring over an ~8-hour period, along with its associated strain-rate front propagation velocity of 2.5 m/hour (Fig. S4)."*

- Lines 315-316: "Our observations are consistent with earlier studies that demonstrate the influence of seasonal precipitation on slope movement". There is a lot of monitoring work done on this landslide using different types of geophysical techniques (e.g. ERT and seismic monitoring). However, here you only compare it with the ShapeArray data. This section should include a bit more (general) discussion on what these other geophysical techniques show that is relevant to your study, and how your results provide more/complementary information about the landslide processes. I don't suggest a detailed comparison as what you have done with the ShapeArray data, but a description of what other works show and how they compare to your observations would be granted here.

Response: Thank you for your suggestion. We have revised this section to incorporate further details of other studies.

Applied changes:

- As per above.

- Lines 338-339: "Future research should investigate the effect of cable geometry on the detectable strain thresholds.". I agree. Along the same lines, how about the effect of cable construction on the measured/detectable strain magnitudes? I know that you only had one type of cable, but can you (very briefly) comment on how using loose-tube, gel-filled cable, for example, would affect strain measurements? A brief mention is sufficient since it is beyond the scope of your paper to compare cables, but I think it is an important consideration when planning deployments for low-frequency DAS strain monitoring and it is worth mentioning. You could also discuss this after the discussion on strain transfer in comparison with ShapeArray ending in line 361).

Response: Thank you for your suggestion. We would like to clarify the above, as there was also a loose-tube cable installed. However, analysis and comparison of the two cable types was outside of the scope of this paper. We have added details on this in the introduction.

Applied changes:

- L123-127: *"Most of the fiber-optic cable installed is of tight-buffered construction, but 140-m of gel-buffered cable was placed alongside a similar length of the tight-buffered cable for comparison. Our analysis focuses exclusively on data acquired from the tight-buffered cable due to its improved strain transfer properties. The British Geological Survey acquired data along the entire cable length using ..."*
 - L443-453: *"Further to the above, different types of cable construction will result in different strain transfer coefficients. Our study uses DAS results acquired from a tight-buffered fiber optic cable, providing improved strain transfer from the surrounding formation to the cable. In contrast, a loose-tube (gel-filled) cable type is designed to minimize the strain transferred to the optical fiber from the surrounding formation. As such, this cable type is deemed less effective for monitoring changes in strain⁴⁷. However, this same attribute increases the effectiveness of loose-tube cables for temperature monitoring. Although our study focuses on strain changes in the near-surface for landslide characterization using DAS, complementary DFOS technologies such as distributed temperature sensing could be considered to provide measurements of temperature changes along the fiber and support interpretation of DAS data."*
- Throughout the discussion, I think it is worth making clear that this is a shallow landslide and movements are in the near-surface, where the cable is deployed. The methods could be less effective for deep-seated landslides where the movements happen at depth.

Response: Thank you for your suggestion. We have made revisions throughout the discussion to clarify this is a shallow landslide and the cable is deployed in the near surface. We also include our original statements below that are included in our discussion for completeness.

Applied changes:

- L369-370: (original): *"This is likely a result of the main scarp providing a direct rainfall-infiltration pathway to the near-surface cable"*
- L380-382 (original): *"This is likely a result of our interpreted slip-surface geometry, where the greatest observed tensile strains at the near-surface cable are expected to occur where the slip surface intersects with the cable."*
- L387-390: *"Additional observed strain-rate patterns are likely impacted by a combination of factors, including slope topography, strain redistribution from the shallow landslide movements, and variations in material parameters"*
- L415-419: *"For example, acquiring DAS data from a downhole fiber optic cable paired with an inclinometer or ShapeArray, combined with a trench installation would provide useful information on the capabilities of DAS to capture deformations occurring at depth. Regardless, our experiment demonstrates the valuable insights gleaned from a near-surface trench installation."*

Materials and Methods

- How long are the data files recorded by the IU? Do you concatenate the data for the entire period of interest before you conduct your decimation, or do you apply your decimation file by file? Are steps 6 and 7 applied to individual files or a long file encompassing the entire recording period? Please clarify.

Response: Thank you for your questions. Answers are as follows:

- The data files recorded by the interrogator are of unknown time length. The data was shared by OptaSense using a proprietary format (.cvt) containing 32 files of 10.7 GB individual file sizes). The data was processed in OptaSense DxS software to decimate from 500 Hz to 50 Hz and were exported to the HDF5 format for individual cable segments with one-minute duration file lengths.
- The HDF5 one-minute files were decimated from 50 Hz to 1 Hz using SciPy (Python)

- The resulting data arrays are then concatenated over the entire period of interest prior to converting the optical phase data to strain data via Equation 1. As such, the final steps of implementing the median filter and integration are applied to a long file encompassing the entire recording period.

We agree these are important points to include in our description of data processing steps and have updated this section accordingly.

Applied changes:

- L562-580: *DAS data are acquired using the OptaSense ODH-F interrogator unit in the quantitative mode. The obtained data represent the optical phase change at a sampling rate of 500 Hz. Data are acquired from OptaSense in a proprietary file format of 32 files, with individual file sizes of 10.7 gigabytes. They are decimated to 50 Hz with a low-pass anti-aliasing filter and exported to HDF5 file format using the OptaSense DxS software, where each file contains the channels corresponding to a cable segment (labelled L1 to L6 from west to east) of one-minute duration.*
- *The remainder of the data processing steps are completed using the following open-source Python modules: SciPy⁶⁴, NumPy⁶⁵, Pandas, H5Py, Obspy, Zarr and Xarray⁶⁶. The processed strain and strain-rate DAS datasets are saved using Xarrays containing appropriate metadata (time, channel and coordinates) in NetCDF files corresponding to individual cable segments (six segments numbered L1 to L6 from west to east) with explanatory Jupyter notebooks detailing the following processing steps⁶⁷:*
 1. *The DAS data are decimated to 1 Hz following the application of a low-pass antialiasing filter.*
 2. *Following decimation, the data arrays are reshaped, concatenating the one-minute files to incorporate the full data acquisition time period (approximately three days).*
 3. *The concatenated optical phase data are converted to strain using Equation 1.*

- Line 458-460: “Short periods of data gaps...” - DAS measures relative strain change, and stopping and restarting the measurements causes losing your reference strain state, sometimes. Can you comment on how large these data gaps are and how you deal with them in your integration? How do you make sure that you are not seeing a change in strain at each data gap?

Response: Thank you for pointing out this important point for consideration. We have modified this section for clarity by creating a section header and adding appropriate details. Following review, we confirm that these data gaps are not attributed to power-related issues of the DAS interrogator. Indeed, as you point out, stopping and restarting the measurements will cause losing the reference strain state. We have added an additional statement in the Background to reflect this.

Applied changes:

- L164-165: *"The optical phase changes and corresponding strain measurements represent a change from a baseline measurement occurring at the start of data acquisition."*
- L595-599: **"Data gaps:** *Three periods of data gaps, approximately 45-minutes duration, are visible in Figures 2, 3 and 5 are due to corrupted data files, where the original data was unable to be recovered. These periods are excluded from the above data processing steps. The data is reindexed to a continuous time vector using the timestamp associated with each filename, and the data gaps are incorporated as 'NAN' (i.e., not a number) values."*

- Line 485: “)” needs to be removed.

Response: We have updated this to remove the “)”

Applied changes:

- L497: Revised to *"(Equation 1)"*

- I appreciate the effort to calculate the effect that temperature changes affecting the cable can have on strain measurements. Did you observe any effects associated with temperature changes affecting the IU itself? They should manifest as “common-mode noise” along the cable. Did you have any temperature control in the barn?

Response: No, we did not observe any effects associated with temperature changes affecting the IU. There was no temperature control in the barn.

Applied changes:

- N/A

- Another way of monitoring temperature changes along the fiber would be to deploy DTS along the same cable (different fiber). I suggest mentioning this in the section describing the temperature effect evaluation, or even in the Discussion section of the paper, as an additional consideration for future experiments of this type.

Response: We agree with your suggestion and have added a paragraph to our Discussion on the deployment of DTS, as per the below.

Applied changes:

- L450-453: *"Although our study focuses on strain changes in the near-surface for landslide characterization using DAS, complementary distributed fiber optic sensing technologies such as distributed temperature sensing could be considered to provide measurements of temperature changes along the fiber and support interpretation of the DAS dataset."*

General Minor suggestions

- I suggest mentioning the name of the study site already in the Abstract. It is mentioned for the first time in the Introduction with not much context (lines 78-79), and then directly introduced in the Background section.

Response: Thank you for your suggestion. We have updated our abstract to mention the name of the study site.

Applied changes:

- L27 (Abstract): Updated statement as follows, *"We employ distributed acoustic sensing (DAS) strain data below 1 Hertz frequency over a three-day period of rainfall at the Hollin Hill landslide, and quantify strain-rate changes at meter and sub-minute scales."*

- Why is the photo of fiber installation the very last one in the Supplementary (Fig. S6) if it is the first one to be mentioned in the text? I would suggest moving it to be the first one.

Response: Thank you for your suggestion. We have reorganized the Supplementary figures such that their order follows their mention in the text.

Applied changes:

- Supplementary Figures: Reorganized all figures such that they follow the order they are mentioned in text.

- Line 146: “.. n corresponds to a refractive index” - I would be a bit more specific and say “refractive index of the fiber”

Response: Thank you for your suggestion. We have modified this statement accordingly.

Applied changes:

- L162: "...where n corresponds to a refractive index of the fiber ,"

- Line 409-410 : " Selecting a channel spacing less than the LG is akin to applying a moving average filter to the dataset" - I think this sentence is a bit confusing and could be rephrased. The gauge length is what acts as a moving average filter. So, I would rather say that when channel spacing is smaller than the gauge length, the effect of the gauge length is the same as applying a moving average to the data, moving one channel at a time (or something along these lines).

Response: Thank you for your suggestion. We have modified this statement to improve clarity.

Applied changes:

- L502-504: Updated sentence as follows: "*When the channel spacing is less than the L_G , the effect of the latter is the same as applying a moving average window of length L_G at each channel.*"

- Figure 3: the time labels seem to be displaced with respect to the tick marks, which makes it a bit confusing.

Response: We have revised the time labels on this figure for clarity.

Applied changes:

- Figure 3 (now Figure 4). Adjusted time labels to align with tick marks.

- Figure 5: is there a data gap In the ShapeArray data on the West lobe? I think it is not mentioned in the text.

Response: Although we mention the data gap in our Materials and Methods, we appreciate your comment and have included an additional mention in the Figure 5 caption for clarity.

Applied changes:

- Former Figure 5 (now Figure 6). We added the following statement to the caption: "*The west lobe ShapeArray did not record data from January 14 at 11:00 to January 15 at 13:00 for reasons unknown.*"

- Figure 5: it might be useful to plot precipitation and soil moisture on top of the displacements, for completeness and aid reference to the previous figures and observations.

Response: Thank you for your suggestion. We have revised this figure to include precipitation and moisture data.

Applied changes:

- Revised Figure 5 (now Figure 6) to include precipitation and moisture data.

REVIEWERS' COMMENTS

Reviewer #2 (Remarks to the Author):

Dear authors,

Thank you for your work on the manuscript review. I think that the article is now much clearer and highlights the novelty and contributions of this study to the understanding of processes in the Hollin Hill landslide, and also to the use of low-frequency DAS for deformation monitoring.

Most of my comments have been properly addressed, and the authors have done a great job improving the clarity of the results and the discussion. I appreciate the addition of Figure 3, explaining the conceptual framework followed by the authors to interpret the strain and strain-rate measurements in terms of landslide processes.

I think the paper is of good quality and deserves publication after conducting some minor clarifications and corrections that I have detailed in the attached file.

Once again, thank you for this interesting contribution and for your hard work on the review.

[Editorial note: Please find reviewer #2's attached document beginning on the next page.]

Minor comments to “Previously hidden landslide processes revealed using distributed acoustic sensing with nanostrain-rate sensitivity”, by S. Oullet et al. (R1)

- Lines 75-76: “...demonstrate the capability of low-frequency DAS to measure changes in strain at the Hollin Hill Landslide Observatory.” - I still think that this is a bit of a “sudden” mention of the site. I think you could remove the mention to Hollin Hill here and say “ ... demonstrate the capability of low-frequency DAS to measure changes in strain associated to the movement of slow-moving shallow landslides”. Then, later in line 80, you can add: “... the kinematics of the slow-moving landslide of the Hollin Hill Landslide Observatory in England.”
- Line 31: “... are like previously observed...” - I would change it to “agree with previously observed...” or “correlate with previously observed...”
- Line 94: Do you mention somewhere the geographical location of Hollin Hill? This information should be added here.
- Lines 115-116: “describes zones of materials with different relative densities, leading to accumulating strains downslope” - I am not sure what you mean by this sentence. Are you talking about the strain observed in DAS? If so, I would suggest that you don’t mention strain here, as you haven’t talked about the DAS results here yet. I also don’t really understand how “material with different relative densities” lead to “accumulating strain downslope”. Can you clarify this sentence? [This sentence is much clearer in the Discussion... I suggest you rephrase it here, or you remove it since it reads as a comparison with your data, which you haven’t shown yet.]
- Line 121: “...gel-buffered cable...” - I think it should be “loose-tube, gel-filled”.
- Line 144 (Figure 1 caption): I think this “2” should not be there.
- Line 154: “... over the gauge length of the measurement...” - Please add “(L_G)” after “gauge length”, as this symbol appears in Equation 1 but I think it is not defined.
- Line 196-197 (Figure 2 caption): “DAS channel locations featured in Fig.4C (upper solid line) and Fig.4D (lower solid line)” - I think it should be the other way around. The channel in Figure 4C shows a decrease in strain (lower solid line) and the channel in Figure 4D shows an increase in strain near the scarp (upper solid line).
- Line 199 (Figure 2 caption)- Figure 4A should be Figure 4B.
- Line 202: “Due to the spatiotemporal characteristics...” - Do you mean “Thanks to...”? It would work better, I think.
- Line 246: “Figure 4c” - I guess it should be “4C” to be consistent with your capitalization of the letters as in all other figure references.
- Line 256-257: “This coincides with accelerating strain near the main scarp (Fig 4D).” - You could also add a reference to Figure 2C here, upper solid line, since that is where the channel shown in 4D is taken from.
- Line 268-270: Fig. 4A should be “Fig 4B”.

- Lines 305-306: “maximum strain changes in the rupture zone to the west of the existing scarp” - Based on Figure 5, should this be “to the southwest of the existing scarp”?
- Line 315: “Figure 5” should be “Figure 6”.
- Line 350 - “...as it demonstrates the scale invariance of the earlier observed landslide processes because of the extreme sensitivity of our DAS-based method” - This sentence could be a bit clearer. What do you exactly mean by “scale invariance”? Are you referring to improved spatial and temporal resolution? Or also to sensitivity of small movements (since you mention the “sensitivity” here as well)? I would make this sentence a bit more explicit and say that your work reveals high variability of landslides processes both in space and time which could not be resolved by previous measurements as they had limited spatial and temporal resolution and/or limited sensitivity to small movements (or something along these lines).
- Line 362-363: “this addresses existing limitations of state-of-the-art remote sensing technologies such as ground-based interferometry and Doppler radar” - I would suggest to add “...addresses existing limitations of current deformation monitoring techniques based on state-of-the-art remote sensing...”, to emphasize that you are talking about the tools that are currently used to monitor movement on landslides.
- Lines 379-380: “...This is likely a result of our interpreted slip-surface geometry at this location...” - A reference to Figure 3A might be useful here.
- Lines 436-439: The difference in magnitude can also be due to the fact that the ShapeArray is a point measurements and DAS measures strain-rate over the gauge length, which acts as an averaging filter. You could add a sentence on this effect too, as an additional consideration.
- Lines 448-449: “However, this same attribute increases the effectiveness of loose-tube cables for temperature monitoring” - I thought that tight-buffered cables are also more sensitive to temperature changes than loose-tube, gel-filled cables?
- Lines 503-504: “...average window of length L_G at each channel.” - I would maybe say “... of length L_G centered at each channel.”
- Line 626: The bracket should be removed.

Reply to peer reviewer comments

Reviewer #2 (Remarks to the Author):

Dear authors,

Thank you for your work on the manuscript review. I think that the article is now much clearer and highlights the novelty and contributions of this study to the understanding of processes in the Hollin Hill landslide, and also to the use of low-frequency DAS for deformation monitoring.

Most of my comments have been properly addressed, and the authors have done a great job improving the clarity of the results and the discussion. I appreciate the addition of Figure 3, explaining the conceptual framework followed by the authors to interpret the strain and strain-rate measurements in terms of landslide processes.

I think the paper is of good quality and deserves publication after conducting some minor clarifications and corrections that I have detailed in the attached file.

Once again, thank you for this interesting contribution and for your hard work on the review.

Response: Thank you for sharing your feedback and for your encouraging review of our submission. We are pleased to hear that we were able to adequately address your comments. We appreciate your thorough review and details of our responses are below, in point-by-point format.

Further to this, we have revised all our figures and labels to meet the editorial requirements of the journal. During the updates to the figures, we realized that there had been an error in the strain axis labels of Figure 4f. Previously, the strain axis labels varied from +1 to -2 millistrain, when they should have been +0.1 to -0.2 millistrain. The strain-rate axis labels are correct and have not been changed. We have revised the strain axis labels in Figure 4f accordingly and have confirmed these did not affect any of our results in the main manuscript text.

L75-76: "...demonstrate the capability of low-frequency DAS to measure changes in strain at the Hollin Hill Landslide Observatory." - I still think that this is a bit of a "sudden" mention of the site. I think you could remove the mention to Hollin Hill here and say "... demonstrate the capability of low-frequency DAS to measure changes in strain associated to the movement of slow-moving shallow landslides". Then, later in line 80, you can add: "... the kinematics of the slow-moving landslide of the Hollin Hill Landslide Observatory in England."

Response: Thank you for your suggestion. We agree with your proposed changes and have reworded this section to improve the introduction of the site. We have also modified the paragraph in L77-80 to improve readability.

Applied changes:

- L72-75: Other studies, referring to a DC-coupled DAS as a distributed Rayleigh sensing system, demonstrate the capability of low-frequency DAS to measure changes in strain associated with the movement of slow-moving shallow landslides.

- L77-80: We reveal the kinematics of a slow-moving landslide at the Hollin Hill Landslide Observatory in England by employing low frequency (<1 Hz) DAS data over a three-day period. The data are acquired with nanostrain-rate sensitivity, 1 Hz temporal sampling and a 4-meter spatial resolution over 925 m of optical fiber.

L31: "... are like previously observed..." - I would change it to "agree with previously observed..." or "correlate with previously observed..."

Response: We agree with your suggested rewording and have updated this sentence per the below ("agree with ...").

Applied changes:

- L31: "However, the millimeter processes over three days agree with previously observed seasonal landslide patterns."

Line 94: Do you mention somewhere the geographical location of Hollin Hill? This information should be added here.

Response: We did not include the geographical location of Hollin Hill. We agree with your suggestion and have added a sentence to include this information.

Applied changes:

- L92: "The Hollin Hill Landslide Observatory, located in North Yorkshire, UK³², is one of the most studied slow-moving landslides in the world and has been monitored by the British Geological Survey since 2008^{3,33}."

Lines 115-116: "describes zones of materials with different relative densities, leading to accumulating strains downslope" - I am not sure what you mean by this sentence. Are you talking about the strain observed in DAS? If so, I would suggest that you don't mention strain here, as you haven't talked about the DAS results here yet. I also don't really understand how "material with different relative densities" lead to "accumulating strain downslope". Can you clarify this sentence? [This sentence is much clearer in the Discussion... I suggest you rephrase it here, or you remove it since it reads as a comparison with your data, which you haven't shown yet.]

Response: Thank you for your comments on this section. This section describes an earlier study and is intended to provide additional background to our work. Considering your feedback and that we have also commented on the relevant findings from this study in our later discussion, we agree with removing this sentence.

Applied changes:

- L112-114: Removed the following sentences from the introduction: "A Hollin Hill landslide model, incorporating seismic refraction tomography and geoelectrical resistivity modelling, describes zones of materials with different relative densities, leading to accumulating strains downslope. Drainage near the base of the flow lobes results in stabilizing material near the toe."

Line 121: "...gel-buffered cable..." - I think it should be "loose-tube, gel-filled".

Response: Thank you for pointing this out, that is correct. We have revised this accordingly.

Applied changes:

- L116: Revised sentence as follows: "Most of the fiber-optic cable installed is of tight-buffered construction, but 140-m of loose-tube, gel-filled cable was placed alongside a similar length of the tight-buffered cable for comparison."

Line 144 (Figure 1 caption): I think this "2" should not be there.

Response: Thank you, you are correct (this was a typo). We have removed this.

Applied changes:

- Figure 1 Caption: Removed standalone "2".

Line 154: "... over the gauge length of the measurement..." - Please add "(L_G)" after "gauge length", as this symbol appears in Equation 1 but I think it is not defined.

Response: Thank you for your suggestion. That is correct, we had defined it later on, but it was not defined beforehand. We have added the gauge length acronym to this sentence.

Applied changes:

- L135: "...Rayleigh-backscattered light over the gauge length (L_G) of the measurement ..."

Line 196-197 (Figure 2 caption): "DAS channel locations featured in Fig.4C (upper solid line) and Fig.4D (lower solid line)" - I think it should be the other way around. The channel in Figure 4C shows a decrease in strain (lower solid line) and the channel in Figure 4D shows an increase in strain near the scarp (upper solid line).

Response: Thank you for pointing this out. We have revised this accordingly.

Applied changes:

- Figure 2 caption: "c Strain-rate for cable section 4, highlighting Sequence 2 and DAS channel locations featured in Fig.4c (lower solid line) and Fig.4d (upper solid line)."

Line 199 (Figure 2 caption)- Figure 4A should be Figure 4B.

Response: Thank you for pointing this out. You are correct, we have revised this accordingly.

Applied changes:

- Figure 2 caption: "e Strain-rate for cable section 6, highlighting Sequence 1 and DAS channel location featured in Fig. 4b."

Line 202: "Due to the spatiotemporal characteristics..." - Do you mean "Thanks to..."? It would work better, I think.

Response: We agree with your suggestion and have updated this accordingly.

Applied changes:

L173: "Thanks to the spatiotemporal characteristics of the DAS data, velocity features of the landslide ..."

Line 246: "Figure 4c" - I guess it should be "4C" to be consistent with your capitalization of the letters as in all other figure references.

Response: Thank you - we have updated all figure labels to conform with the editorial standards and for consistency (lower-case for all).

Applied changes:

Updated all figure labels to conform with editorial standards.

Line 256-257: "This coincides with accelerating strain near the main scarp (Fig 4D)." - You could also add a reference to Figure 2C here, upper solid line, since that is where the channel shown in 4D is taken from.

Response: Thank you - we agree with your suggestion and have updated this accordingly.

Applied changes:

L218: "This coincides with accelerating strain near the main scarp (Fig 4d, Fig 2c, upper solid line)."

Line 268-270: Fig. 4A should be "Fig 4B".

Response: Thank you for pointing this out - we have updated this accordingly.

Applied changes:

L231: "The strain continues to increase at more gradual levels at the main scarp up until the end of the DAS acquisition period (Fig. 4b)."

Lines 305-306: "maximum strain changes in the rupture zone to the west of the existing scarp" - Based on Figure 5, should this be "to the southwest of the existing scarp"?

Response: That is correct - southwest is more precise. We have updated this accordingly.

Applied changes:

L238: " ... considering the maximum strain changes in the rupture zone to the southwest of the existing scarp ..."

Line 315: "Figure 5" should be "Figure 6".

Response: Thank you for pointing this out - we have updated this accordingly.

Applied changes:

L246: "We compare relative displacement data from two vertical ShapeArrays installed near the fiber optic cable to a depth of 2.5 m, at the west flow lobe and east flow lobe in 2013 (Fig. 6)."

Line 350 - "...as it demonstrates the scale invariance of the earlier observed landslide processes because of the extreme sensitivity of our DAS-based method" - This sentence could be a bit clearer. What do you exactly mean by "scale invariance"? Are you referring to improved spatial and temporal resolution? Or also to sensitivity of small movements (since you mention the "sensitivity" here as well)? I would make this sentence a bit more explicit and say that your work reveals high variability of landslides processes both in space and time which could not be resolved by previous measurements as they had limited spatial and temporal resolution and/or limited sensitivity to small movements (or something along these lines).

Response: Thank you for your feedback on this section. We have revised this statement to improve clarity.

Applied changes:

L273-275: "This agreement with earlier studies supports the validity of our findings and suggests that the processes governing landslide behavior are scale invariant (i.e., similar regardless of the scale at which they are observed)."

Line 362-363: "this addresses existing limitations of state-of-the-art remote sensing technologies such as ground-based interferometry and Doppler radar" - I would suggest to add "...addresses existing limitations of current deformation monitoring techniques based on state-of-the-art remote sensing...", to emphasize that you are talking about the tools that are currently used to monitor movement on landslides.

Response: Thank you for your feedback - we agree with your suggestion and have updated accordingly.

Applied changes:

L286-287: "Importantly, this addresses existing limitations of current deformation monitoring techniques based on state-of-the-art remote sensing technologies such as ground-based interferometry and Doppler radar¹⁵ which lack temporal resolution combined with sensitivity to small displacements."

Lines 379-380: "...This is likely a result of our interpreted slip-surface geometry at this location..." - A reference to Figure 3A might be useful here.

Response: Thank you for your suggestion. We agree it would be useful, and have included a reference to Figure 3A.

Applied changes:

L304-307: " This is likely a result of our interpreted slip-surface geometry at this location, where the greatest observed tensile strains at the near-surface cable are expected to occur where the slip surface intersects with the cable (Fig. 3a).

Lines 436-439: The difference in magnitude can also be due to the fact that the ShapeArray is a point measurements and DAS measures strain-rate over the gauge length, which acts as an averaging filter. You could add a sentence on this effect too, as an additional consideration.

Response: That's a great suggestion. We have added a statement in the Results section on "DAS-derived displacements with geotechnical instrumentation" to account for the difference in magnitudes , and then revised this statement to include a consideration for the different nature of the measurements.

Applied changes:

L257-260: "... and (4) the different nature of the measurements, where the ShapeArray data provide a discrete measurement at a sensor and the DAS data provide a distributed measurement over a 4-m length of optical fiber (i.e., the gauge length; L_G)."

L362-366: "This relationship between ground displacement from ShapeArray instrumentation and DAS inferred displacement can be extended to estimate a strain transfer of 0.5, under the assumption of similar coupling throughout the length of installed fiber optic cable and with consideration to the different nature of the measurement (i.e., point-based versus distributed; see Results).

Lines 448-449: "However, this same attribute increases the effectiveness of loose-tube cables for temperature monitoring" - I thought that tight-buffered cables are also more sensitive to temperature changes than loose-tube, gel-filled cables?

Response: Thank you for your comment. We were unable to find any studies providing a direct comparison of tight-buffered and loose-tube cables with DAS for temperature change monitoring. Sidenko et al. (2022) perform two separate experiments, where (1) a gel-filled loose-tube cable is used with DAS/DTS and (2) the response of a tight-buffered cable and a bare fiber are compared. However, they do not compare the loose-tube with the tight-buffered in the same experiment. We have removed this sentence as we do not have references to support this statement and it is not necessary for the discussion.

Applied changes:

L374-376: Removed the following sentence: "However, this same attribute increases the effectiveness of loose-tube cables for temperature monitoring."

Lines 503-504: "...average window of length L_G at each channel." - I would maybe say "... of length L_G centered at each channel."

Response: Thank you for your suggestion. We have updated this sentence accordingly.

Applied changes:

L425-427: "When the channel spacing is less than the L_G , the effect of the latter is the same as applying a moving average window of length L_G centered at each channel.

Line 626: The bracket should be removed.

Response: Thank you for pointing this out. We have corrected this.

Applied changes:

L549-551: "Over the DAS acquisition period, the ShapeArrays acquired data with a varying sample interval between 30-minutes to one hour."